# Exploration of cell state heterogeneity using single-cell proteomics through sensitivity-tailored data-independent acquisition

Valdemaras Petrosius[1], Pedro Aragon-Fernandez [1], Nil Üresin[1,2,3], Gergo Kovacs[1], Teeradon Phlairaharn[1,4,5,6], Benjamin Furtwängler [1,2,3], Jeff Op De Beeck[7], Sarah L. Skovbakke [1], Steffen Goletz [1], Simon Francis Thomsen [8], Ulrich auf dem Keller[1], Kedar N. Natarajan [1], Bo T. Porse [2,3,9] & Erwin M. Schoof [1] ✉

Single-cell resolution analysis of complex biological tissues is fundamental to capture cell-state heterogeneity and distinct cellular signaling patterns that remain obscured with population-based techniques. The limited amount of material encapsulated in a single cell however, raises significant technical challenges to molecular profiling. Due to extensive optimization efforts, single-cell proteomics by Mass Spectrometry (scp-MS) has emerged as a powerful tool to facilitate proteome profiling from ultra-low amounts of input, although further development is needed to realize its full potential. To this end, we carry out comprehensive analysis of orbitrap-based data-independent acquisition (DIA) for limited material proteomics. Notably, we find a fundamental difference between optimal DIA methods for high- and low-load samples. We further improve our low-input DIA method by relying on high-resolution MS1 quantification, thus enhancing sensitivity by more efficiently utilizing available mass analyzer time. With our ultra-low input tailored DIA method, we are able to accommodate long injection times and high resolution, while keeping the scan cycle time low enough to ensure robust quantification. Finally, we demonstrate the capability of our approach by profiling mouse embryonic stem cell culture conditions, showcasing heterogeneity in global proteomes and highlighting distinct differences in key metabolic enzyme expression in distinct cell subclusters.

Analytical techniques with single-cell resolution are becoming indispensable tools to study complex biological systems. Although invaluable, the aggregated view obtained by bulk cell population experiments is not sufficient to achieve fundamental understanding of human development and disease. The means to interrogate the first two aspects of the central dogma of biology (DNA-RNA-Protein) are well established and have been widely adopted, but the study of proteomes by liquid chromatography coupled mass spectrometry (LC−MS) at single-cell resolution is just entering the biological application phase[1]. It is estimated that a single mammalian cell contains 50−450 pg of protein[2], posing significant challenges to protein identification and quantification. However, these challenges are to a large extent being mitigated by advances in different aspects of LC−MS-based proteomics[3–13].

Pioneering studies could quantify hundreds of proteins from a single cell[9,13]. These reports marked an important milestone for mass-

spectrometry based single-cell proteomics (scp-MS), however analysis required long chromatographic gradients, complicating practical implementation of large-scale scp-MS investigations. Data-dependent acquisition (DDA) based methods have dominated the field thus far, led by the development of SCoPE-MS approach[4,10,11,14]. The method utilizes isobaric TMT labeling to multiplex single cells and combines them with a carrier channel containing 100–200 cells, allowing parallel analysis of up to 16 cells in a single run with the latest TMTPro 18-plex reagent set. This tremendously improved the throughput and proteome coverage of scp-MS, but in-depth explorations of the biases introduced by the carrier channel in terms of protein quantification have clarified the benefits and limitations of this method[7,10,15,16]. Latest label-free quantification (LFQ) -based approaches have significantly improved the proteome coverage (1000–2000 proteins) and surpass DDA multiplexing based workflows, although the low throughput remains a significant challenge[12,17]. A dual-column LC configuration has been proposed as a potential solution, but is yet to be demonstrated on actual single-cell input[18]. Data-independent acquisition (DIA)[19,20] based approaches have also been used to tackle single-cell proteomes and currently provide the deepest proteome coverage[3,6,21]. Furthermore, the introduction of plexDIA increased the throughput by allowing single-cell multiplexing, similarly to SCoPE-MS, demonstrating great potential for increased throughput in DIA-based approaches[6].

Due to the ultra-low amount of peptides derived from a single-cell, long injection times (ITs) are required to ensure sufficient ions are collected for identification and quantification[7,11,12,15,22]. This limits the capacity of DDA based methods to comprehensively sequence all the peptides present in the sample, putting great demands on analysis efficiency in terms of effectively using available mass analyzer time[7,23]. In contrast, DIA does not suffer from such limitations as multiple peptides are co-isolated and analyzed, potentially acquiring both the MS1 and MS2 spectra of all the precursor ions present in the samples[24]. However, identification and quantification can be hindered by spectra convolution and low signal intensity. Improvements in chromatographic separation have the potential to benefit all types of scp-MS workflows, by providing higher resolution (sharper peaks boosting peptide ion flux), better separation capacity and more stable retention times run-to-run. Accordingly, narrow-bore columns and perfectly ordered micropillar-array-based nano-HPLC cartridges (µPAC) have been manufactured and have shown promising results for ultra-low (<1 ng) input proteomics[17,25–27]. µPAC columns have shown great promise for low-input (<10 ng) proteomics, with high separation power and exceptionally robust peptide retention times[25,26]. Impressively, the improvements brought about by the µPAC columns allowed quantification of proteins from only 50 pg of input[27].

DIA holds great promise for scp-MS and low-input proteomics, however optimal method designs with regards to input load have not been comprehensively investigated. In this study, we carry out survey experiments to determine to which extent optimal DIA method designs are dependent on the sample input load. We build further on our findings by utilizing a high-resolution MS1 (HRMS1)-based DIA approach, to generate a new low-input DIA method design, which we combine with the newly developed µPAC Neo Low Load analytical column. We showcase that with a combination of advanced data acquisition and latest-generation chromatography, we can obtain proteome coverage from low-input (10 ng) samples that is reminiscent of standard (100 ng) samples. A strong focus throughout this work was on keeping sample throughput high, and therefore we opted to assess short gradients only, as implemented either on an Ultimate3000 with flow rate-ramping, or an EvoSep One chromatography system for the initial DIA scheme evaluations. To align our workflows with other published methods[3,12,28], we carried out analysis of HEK293 and display that our method could capture canonical cell cycle driven variation. We epitomize our study by proteome profiling of mouse embryonic

stem cells (mESC) that are cultured across ground-state and differentiation-permissive culture conditions and highlight proteome expression profiles in distinct cell subclusters with a focus on key metabolic enzymes.

## Results

### Increasing low-input sample proteome coverage by wide DIA isolation windows

Increasing the isolation window size during DIA-based acquisition should in theory hamper peptide identification due to more extensive precursor co-isolation resulting in increasingly chimeric spectra. While this effect is pronounced for high-load (>10 ng) samples, we hypothesized that co-isolation constraints are not as prevalent when handling low-load samples (<10 ng). To test this, we carried out a series of experiments where we injected different amounts of Hela digest (100, 10, 5, and 1 ng) and acquired the MS spectra with DIA methods of varying isolation window sizes and resolutions combined with varying ion ITs, while maintaining approximately the same scan-cycle time (Fig. 1a, Supplementary Data 1). As expected, 100 ng of input material resulted in the highest number of protein identifications. Doubling the isolation window width from 10 to 20 m/z, and doubling the resolution slightly increased the proteome coverage, however further widening beyond 20 m/z had an opposite effect (Fig. 1b). In contrast, when lower amounts of peptide were injected, 40 m/z isolation window gave the best results for 10 and 5 ng. Decreasing the peptide load to 1 ng further moved this optimal value to 80 m/z (Fig. 1b), suggesting that the chimeric spectra effects due to co-isolation at such loads are outweighed by increased resolution and IT that enhance the sensitivity. The chosen scan-cycle was coordinated with the chromatographic method to ensure that enough data points per elution peak were acquired to maintain robust sampling[29]. Varying the active gradient length can affect the peptide elution peak width and the chosen scan-cycle time should be aligned with this timeframe[30]. With our chosen parameters, all the methods had a median of 6 or more points-per-peak ensuring comparable quantitative potential (Fig. 1c). Interestingly, although the scan cycle time was kept constant, increasing the resolution, isolation window size, and ITs, led to more data points-per-peak (Fig. 1c). Accordingly, protein quantification precision also improved as more data points were collected, which was especially marked at the lowest-level 1 ng injections (Supplementary Fig. 1A). The additional points are detected potentially due to longer ITs which allows quantification of the elution profile tails that fall below the background intensity at shorter ITs. Together, these findings indicate that detrimental chimeric spectra effects can be overcome in low-input samples by sufficiently increasing the resolution/ITs, facilitated through wider DIA isolation windows.

### HRMS1-DIA in combination with wide isolation windows enhanced quantified proteome depth

Since DIA acquires both MS1- and MS2-level spectra, quantification can be carried out on either level, with the latter commonly being attributed to be more accurate in the literature, as it can overcome co-elution biases[31,32]. Due to this, MS2-based quantification is generally preferred in DIA experiments and is the default output by most popular search engines, such as Spectronaut and DIA-NN[31,33]. A method that breaks away from this convention has also been proposed, termed high-resolution-MS1 (HRMS1) DIA[34–36]. While in standard DIA, the MS1 scan is followed by MS2 scans that sequentially measure the whole m/z range of interest, HRMS1 slices the total m/z range into smaller segments, interjecting MS1 scans in between (Supplementary Fig. 2A). This modification drastically decreases the amount of MS2 data points acquired for each precursor, eliminating the ability to perform robust quantification on the fragment level. Quantification becomes primarily focused on the MS1 information, while the MS2 is used only for identification. Accordingly, the available cycle time can now be more

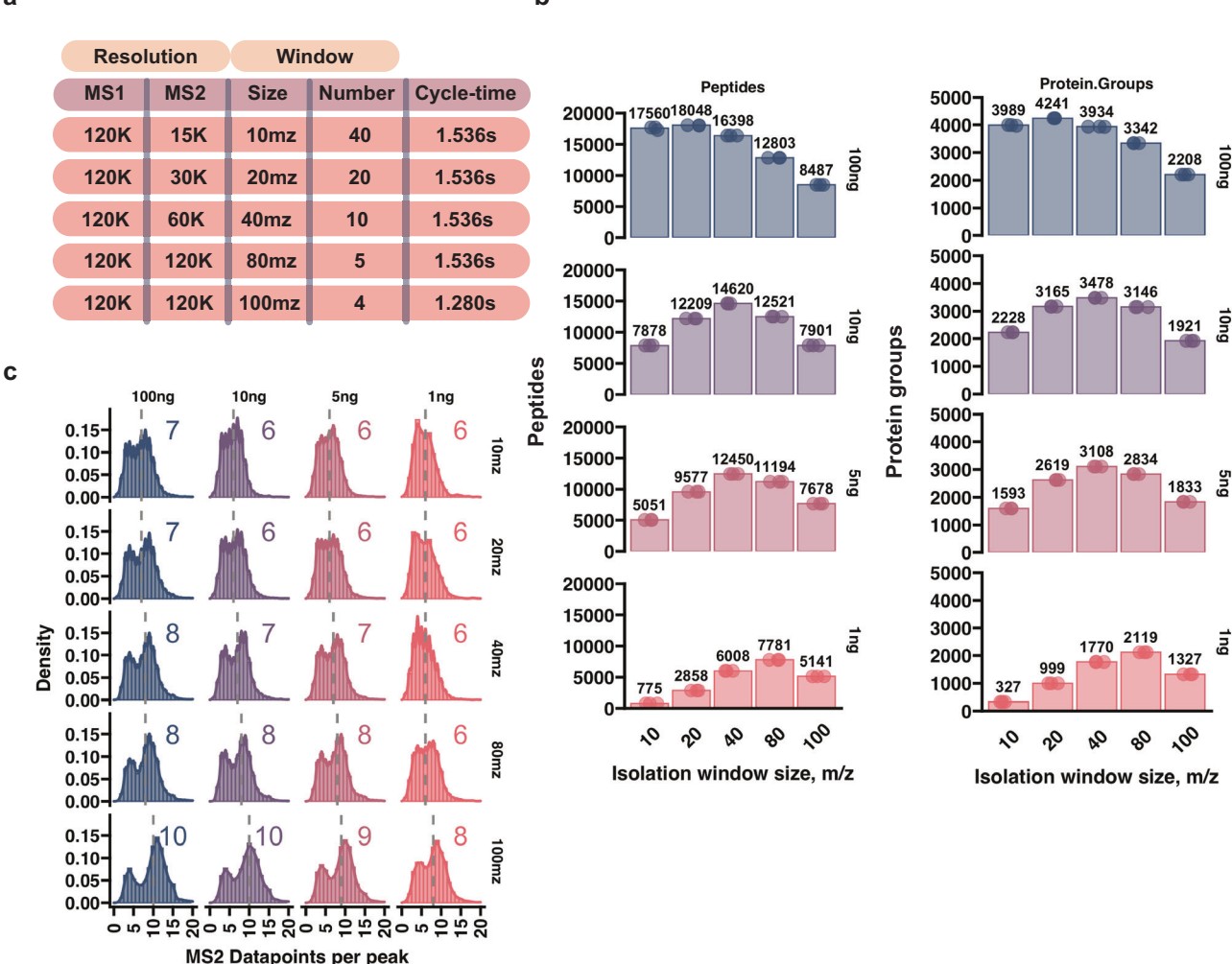

**Fig. 1 | Optimal DIA window isolation size is dependent on the amount of input material. a** Table summarizing the different DIA acquisition methods used. **b** Barplot of peptide and protein quantification numbers with DIA methods that have varying isolation window size and resolution. Numbers above the bars indicate the mean identified protein group number **c** Histograms of points-per-peak (PPP) quantified for peptides. Only one replicate out of three is shown. Gray dashed line marks the median PPP, the number of which is listed at the top of each to the histogram. The Evosep 40SPD whisper method was used for all experiments. Source data are provided as a Source Data file.

optimally used for a segmented part of the overall m/z range, affording longer ITs and higher resolution (Supplementary Fig. 2A). We compared standard DIA versus HRMS1 to determine if we can further increase our proteome coverage with this method. By modifying the DIA acquisition method according to HRMS1, we could increase our resolution (and corresponding ITs) from 30 to 60 K and decrease our isolation window size from 15 to 8 m/z, while maintaining identical scan cycle-times. Not only did HRMS1 significantly outperform standard DIA in terms of identification (Fig. 2a), it also collected more points-per-peak (Fig. 2b) which translated into higher quantitative precision (Fig. 2c). The extra identifications by HRMS1 primarily arose from low-abundant proteins (Supplementary Fig. 2B). We also adopted this modification to linear ion trap (LIT) based DIA[37–39] and observed similar overall performance gains (Supplementary Fig. 3A–C), although it did not surpass OT-based HRMS1-DIA.

We performed a similar isolation window survey experiment as above to see if we could synergize the HRMS1 method with wide isolation windows. In line with our initial observations, widening the isolation window to accommodate for longer ITs and higher resolution scans on 1 ng injections resulted in increased numbers of quantified proteins (Fig. 2d). The protein count peaked at 40 m/z isolation width and decreased once 100 m/z was reached. We term our tailored low-

input method WISH-DIA (**W**ide **Is**olation window **H**igh-resolution MS1-DIA), to encapsulate the combination of wide isolation windows and use of HRMS1 quantification.

Although WISH-DIA showed great promise, the question of quantitative bias remained due to MS1-based quantification. To evaluate this aspect, we utilized a SILAC approach and mixed peptides derived from Hela cells cultured in light or heavy media in different ratios and analyzed the data with the best performing methods (Fig. 2e). While keeping the total sample load to only 1 ng to carefully mimic a low sample-load setting, we directly compared protein abundance (L/H) ratios derived from DIA fragment level or HRMS1 precursor-level (Fig. 2f). Both showed a ratio distribution that was in line with the expected values. There was a clear drop in accuracy as the ratio of heavy and light peptides was increasing, potentially, due to the decreasing proportion of light peptides in the samples making them harder to quantify. MS1 yielded sharper peaks compared to MS2, indicating higher quantitative accuracy, albeit a minor, but clear bias could be observed when 1:1 and 1:2 mixtures were compared on MS1 level quantification, which was not present when MS2 was used (Supplementary Fig. 3D). Interestingly, when higher ratio mixtures were compared, there appeared a minor, but clear discrepancy in MS2-level quantification, while MS1 ratio distribution remained centered around

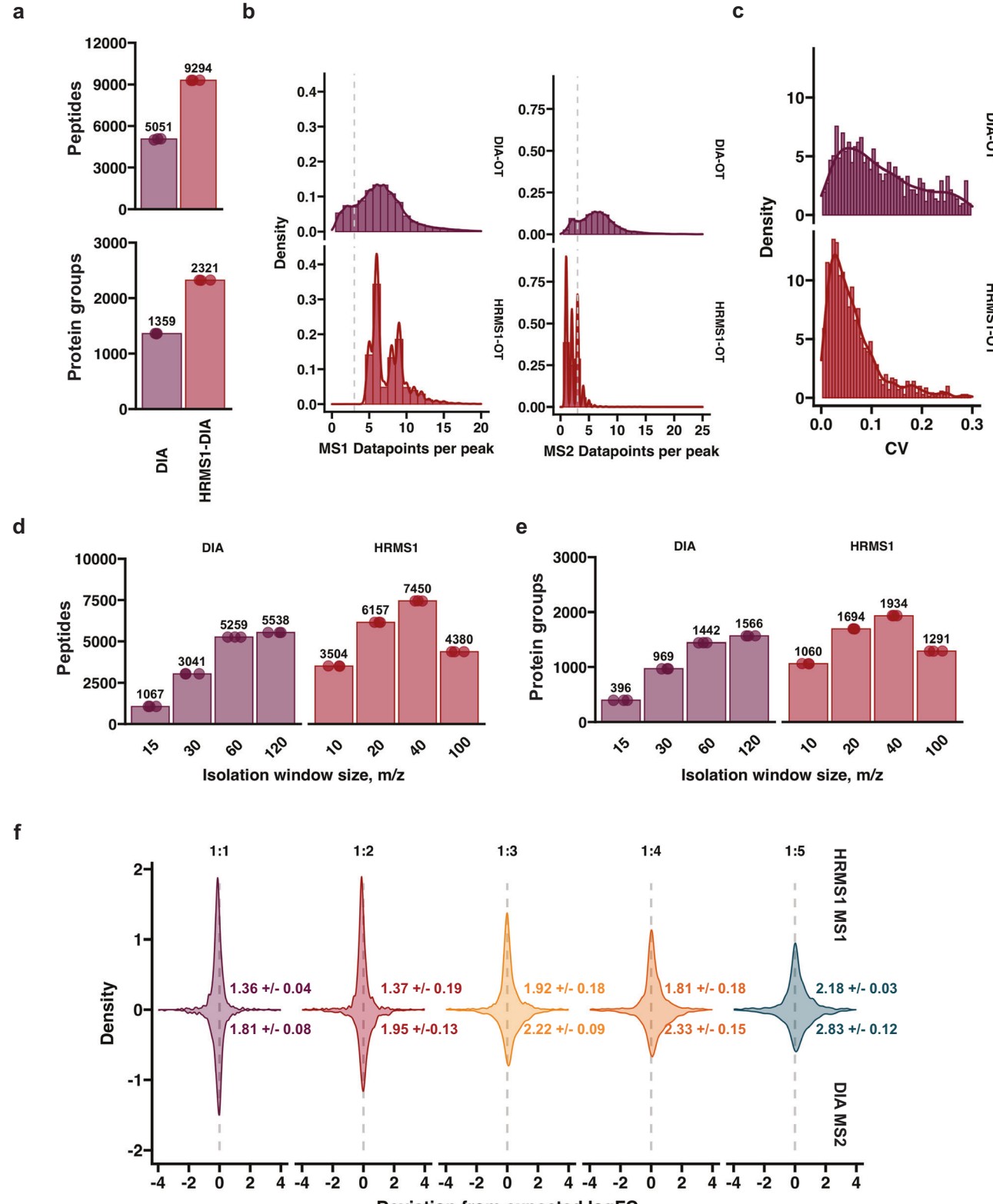

**Fig. 2 | WISH-DIA enables deeper proteome profiling from low-input.**
**a** Quantified number of peptides and proteins with standard DIA and HRMS1-DIA with 1 ng injection with 20PSD Evosep whisper method. Number above the bar indicated the mean identified peptide and protein group number. **b** Acquired MS1 and MS2 points-per-peak. Standard DIA (OT-DIA) is shown in the top and HRMS1-DIA (OT-HRMS1) in the bottom. Gray dashed line indicates a 3 point-per-peak cut off (only for visualization purposes). **c** CV distribution histograms of quantitative precision of standard and HRMS1-DIA. Respectively, MS2 and MS1-based quantification was used for CV comparison. **d**, **e** Barplots of quantified peptides and proteins in wide window HRMS1-DIA acquisition survey with 40SPD Evosep whisper method with 1 ng injection. **f** Heavy and light protein abundance ratio density plots. 1 ng total input material was kept constant and the amount of light and heavy peptides were varied to achieve the required ratios. HRMS1 MS1-based quantification is shown in the top, and DIA MS2-based quantification in the negative side of the plot. Gray dashed line marks the zero values where the measured and expected fold-change matches perfectly. Source data are provided as a Source Data file.

the expected value (Supplementary Fig. 3D). Higher MS1 accuracy for the larger ratios was also observed comparing MS1 and MS2 protein ratios from the standard DIA method (Supplementary Fig. 3E). To provide a more quantitative accuracy comparison we evaluated the quantification error distribution widths (Fig. 2f). We could note that MS1 quantification leads to narrower distribution compared to MS2, for such low-input samples. Taken together, we conclude that WISH-DIA enhances proteome depth from low-input samples while maintaining robust quantitative accuracy.

### Micropillar-array-based nano-HPLC cartridges/columns for low-input proteomics

Next, we substituted the packed C18-beads column with a next-generation µPAC Neo Low-Load column to further augment our low-input workflow efforts (Supplementary Fig. 4A) and explore the general applicability of a WISH-DIA scheme across different chromatography platforms. This 50 cm column has a reduced cylindrical pillar diameter of 2.5 µm, an interpillar distance of 1.25 µm, a total column volume of 1.5 µL, and is non-porous, thereby increasing its chromatographic performance at much reduced loading capacities. We designed methods that utilized flow-ramping up to 500 nl/min to minimize the overhead time needed for peptide break-through and analytical column regeneration (Supplementary Fig. 4B). We generated three single-column and two pre-column configuration methods and tested chromatographic performance of the column by running tryptic digests with our developed WISH-DIA methods (Supplementary Fig. 4C). Examining the peak width of the single-column configuration, we saw that the full-width at half maximum (FWHM) of the peptide precursors peaks is approximately 6.6 second, which broadened to 8.58 seconds for the longest method in line with total gradient time (Supplementary Fig. 4D). Addition of a pre-column in-line resulted in increased peak-widths >9 s, however extending the gradient only resulted in a marginal increase in peak width (Supplementary Fig. 4D). Retention times were very robust and centered across runs, with most precursor elution apex deviations being limited to 2.5 seconds and (Supplementary Fig. 4E). To put the performance into perspective, we compared RT stability with our initially used column and observed a significant reduces RT fluctuations (Supplementary Fig. 4F), underlining the solid chromatographic performance of the µPAC Neo Low Load column. We proceeded to further benchmark the analytical column in terms of proteome coverage for variable amounts of input material.

### Utilizing the synergy between µPAC Neo Low Load and wide isolation window HRMS1-DIA for low-input proteomics

To date, the vast majority of low or ultra-low level input (≤250 pg) studies have focused on DDA based acquisition. It is now possible to routinely quantify >1000 protein groups from such amounts[12,26,28,40,41]. However, this tends to require long LC–MS instrument run-times (>1 h), unless a double-barrel approach is used[18]. First, to try and maximize sample throughput, we evaluated the performance of 45, 26 and 20 minute methods (32, 55 and 72 samples per day (SPD) respectively) and injected different amounts of digested peptide in a single-column configuration (Supplementary Fig. 5A). Commercially available Pierce Hela digest was used (Part #88328), to ensure that our reported performance numbers can be easily evaluated by others. To fully realize the potential of the µPAC Neo Low Load column, we utilized WISH-DIA to quantify proteomes from low-input material (≤10 ng). Optimal methods were identified for each gradient length by carrying out similar isolation window experiments as previously described (Figs. 1–2, Supplementary Data 2) and the best performing methods for all configurations and inputs are summarized in Supplementary Fig. 5A. From 10 ng we quantified from 3000 to 4700 protein groups depending on the method used (Fig. 3a). Decreasing the amount of input material resulted in fewer protein identifications,

albeit up to ~4000 and ~3000 protein groups could still be quantified from 5 and 1 ng respectively. At ultra-low-input level of 250 pg, we quantified 2089 protein groups on average at 32SPD and 1461 at 72 SPD. Overall, our workflow quantifies PG numbers comparable to previously published work, however at 2–3 times greater throughput[17,18,27,42].

To process biologically relevant samples where standard solid-phase extraction[43] cannot be used, a pre-column can be used to ensure robustness on the chromatographic system, and prevent clogging by non-protein contaminants present in the samples. This is especially relevant in single-cell proteomics[1] where indeed prior sample clean-up is not possible. With a tailor-made µPAC pre-column setup, consisting of non-porous 5µm pillars based on C8, we developed 32- and 52-min methods that could quantify similar peptide and protein group numbers as a single-column setup (Supplementary Fig. 5B). Due to the larger sample loop used (20ul vs. 1ul in the single-column setup), the pre-column configuration adds 7 min overhead time to each method, decreasing throughput to 40 and 24 SPD (Supplementary Fig. 4C). With the pre-column configuration, we achieved reminiscent proteome coverage compared to the single-column set-up, where we could quantify >2000 protein groups from ultra-low input (Supplementary Fig. 4B). This was slightly unexpected as the pre-column leads to peak broadening (Supplementary Fig. 4D). As the ultimate goal of our work was to be able to analyze single-cell proteomes, both with high proteome depth and quantitative accuracy, and at reasonable throughput, we next evaluated the performance of WISH-DIA on actual single cells. HEK293 cells were prepared in 384-well Eppendorf low-bind plates with previously described protocols (See "Methods") and transferred to a 96-well plate for injection. Since single-cell samples have been shown to require high ITs[7,11,12,15,22], to accommodate this we further increased the IT and resolution of our WISH-DIA method from 120k (246 ms IT) to 240k (502 ms IT), while doubling the isolation window size (68 m/z) to maintain the same scan cycle time and increasing the resulting proteome coverage (Supplementary Fig. 5C). We processed 10 single cells with our two established 29 min and 52 min pre-column methods and could quantify 717 and 1008 proteins by directDIA (Fig. 3c). However, as also recently shown by others[28], transferring single-cell samples leads to severe signal losses. To test the extent of this effect in our experimental setup, we switched to direct injection of single-cell peptides from their original 384-well plate. Accordingly, direct injection boosted our average identifications by ~60% for the shorter and ~30% for the longer method (Fig. 3c), bringing our quantified protein numbers to 1151 and 1318 when searched with directDIA, which is highly comparable to coverage obtained with low-input specialized instruments (Supplementary Fig. 5E). Quantification robustness was ensured by keeping the cycle time sufficiently short to collect a minimum of 5 data points per precursor elution profile (Fig. 3d), while MS2 data points were only collected for identification (Fig. 3e).

### Quantification quality of additional proteins gained by high-load library use

Some studies have chosen to utilize enhanced search strategies by including higher load libraries (e.g. 10 ng), which can drastically boost the number of quantified proteins. So far, either diluted bulk cell population digests or samples containing multiple cells have been used for this purpose[3,28,44,45]. However, the exact impact of using such high-load (HL) ID transfer approaches remains unclear, especially in terms of quantification accuracy and consequently, biological information captured by the additional proteome coverage. A gas-phase-fractionated library (GPF)[46,47] is another approach that can be used to gain identifications, which is generated by dividing our m/z range of interest into 6 segments of 100 m/z and analyzing samples while acquiring spectra for only that segment (See "Methods"). Due to the decreased m/z range for each individual run, we could therefore

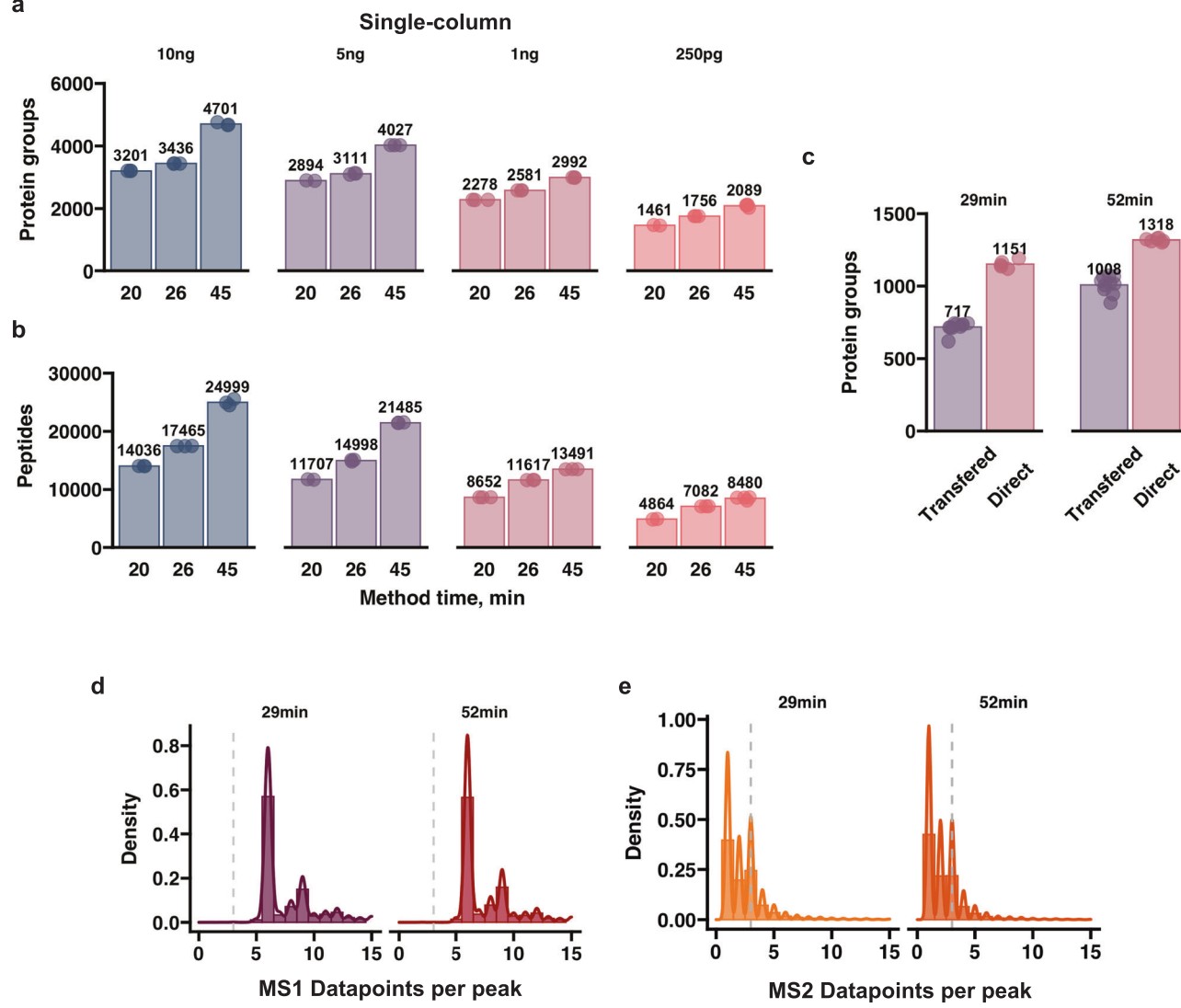

**Fig. 3 | Deep proteome coverage of low-input samples by advances in multiple aspects of mass spectrometry. a**, **b** Barplots of quantified proteins and peptides width different method lengths and peptide loads with the single-column µPAC Neo Low Load set-up. **c** Cells directly injected from a 384-well plate (Direct) or transferred to a 96-well beforehand (Transferred) with two gradient lengths, 240k resolution method with the pre-column configuration. **d**, **e** Histograms of data points per peak on MS1 and MS2 level with the gradient lengths. Gray dashed line indicates a 3 point-per-peak cut off (only for visualization purposes). All reported numbers are obtained with directDIA by searching the runs from the same method in a single batch. Source data are provided as a Source Data file.

further increase our ITs (1014 ms) and decrease the isolation window width, allowing the identification of peptides that have very low abundance and are difficult to quantify in our global WISH-DIA runs.

To assess the protein quantification quality of both approaches, we mixed light and heavy peptides in three different ratios while maintaining a constant 10 ng injection load. We then diluted our sample to 1 ng injections that were used as actual runs and GPF library creation, and the 10 ng were used to acquire HL libraries. To gauge the quantification accuracy we plotted the light and heavy ratio distributions for the identified proteins obtained with directDIA or LibraryDIA with a high-load or GPF library (Fig. 4a). The use of a HL library approximately doubled the coverage, while GPF led to ~50% additionally identified proteins. The enhanced proteome depth was accompanied by substantial widening of the ratio distribution, indicating loss of accuracy in the dataset as a whole (Fig. 4a). To gain a better understanding of how the increased proteome coverage was affecting the overall quantitative accuracy of the data, we extracted the proteins that could be identified with directDIA or only with the implementation of a HL or GPF library and re-plotted the ratio

distributions (Fig. 4b). The additionally quantified proteins of those HL or GPF searches compared to directDIA alonehad a strikingly wider distribution, indicating significantly increased deviation from the true values on those additionally identified peptides and proteins. (Supplementary Fig. 6). As low-abundant proteins are expected to naturally have poorer quantification relative to high abundant ones, we investigated this in greater detail. Application of libraries tremendously improved the identification of proteins in the lowest end of the abundance range (Fig. 4c), but the gained proteins did extend beyond this range. The HL clearly aided the identification of a larger number of proteins found in the lowest end of the abundance range compared to GPF, indicating its higher capacity to extend proteome coverage. Interestingly, the light and heavy peptide ratios were more dispersed throughout the abundance range for both libraries, suggesting that the quantification quality of those gained proteins is potentially rather poor (Fig. 4d). These findings point towards possible challenges with the accuracy of proteins gained via libraries from low-input samples and indicate that extra scrutiny is warranted when biologically interpreting these additional identifications.

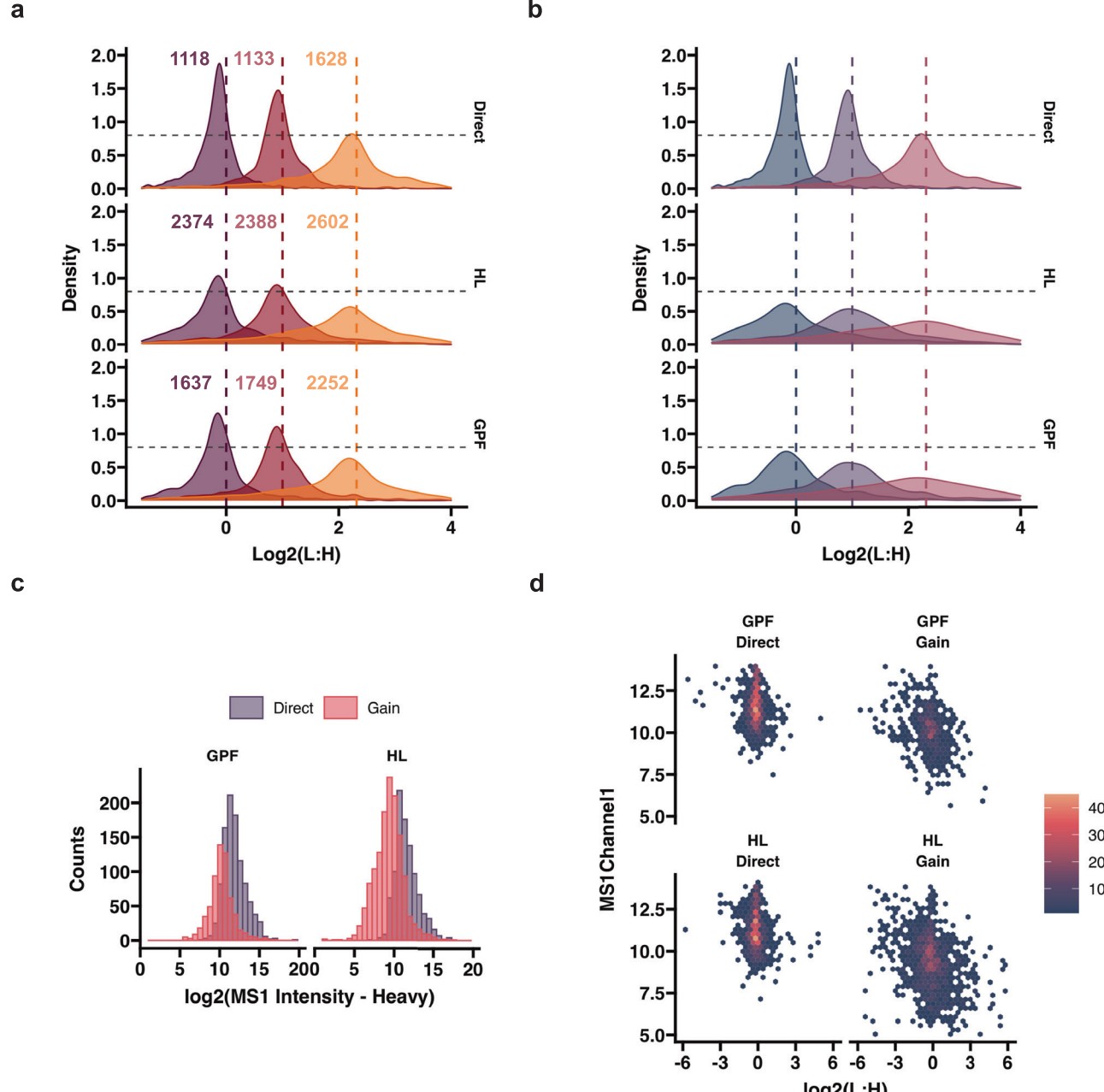

Fig. 4 | Assessing the quality of protein quantification gained by high-load and gas-phase fractionated library based DIA. a Density plots showing the log2 transformed light and heavy protein abundance ratios. Proteins quantified with directDIA shown in top and high-load (HL) library in the middle and gas-phase fractionated (GPF) in the bottom. Dashed lines denote expected ratios and numbers indicate the total number of identified proteins in each mix. MS1-based quantification is used throughout. b Similar to A, but showing the distribution of proteins identified with directDIA and gained with either HL or GPF libraries only (the directDIA found proteins are removed from these datasets). c Histogram showing protein distribution across the log2 transformed abundance range. Only the 1:1 (L:H) mix data is shown. d Hexbin plot showing the distribution of light and heavy (L:H) log2 transformed protein ratios for proteins found with directDIA and gained by the HL and GPF libraries. Source data are provided as a Source Data file.

## WISH-DIA with a next-generation analytical column enables high-quality single-cell proteome profiling

As a proof of concept, we generated a small dataset of 100 HEK293 cells using WISH-DIA in combination with the μPAC Neo Low Load column. We analyzed 62 cells with a 40SPD method and 40 cells with 24 SPD. On average, both methods quantified ~1670 protein groups per cell (Fig. 5a). Although protein quantification was almost identical, the longer method could detect more peptides (Fig. 5a). As an alternative to high-load libraries, we instead opted to generate a gas-phase-fractionated library (GPF[46,47]), by dividing our m/z range of interested into 6 segments of 100 m/z and running single-cell samples while acquiring spectra for only one segment at a time (See "Methods"). Due to the decreased m/z range for each individual run, we could therefore further increase our ITs (1014 ms) and decrease the isolation window width, allowing the identification of peptides that have very low abundance and are difficult to quantify in our global WISH-DIA runs. By applying such a GPF approach to our single-cell runs, we were able to boost our quantified proteins by ~20% (Fig. 5a). As expected, the quantification of these additionally identified proteins was noisier, and primarily spanned the lower range of the abundance distribution (Fig. 5b). All the runs showed a relatively low level of missing values on the protein level, with the vast majority of cells

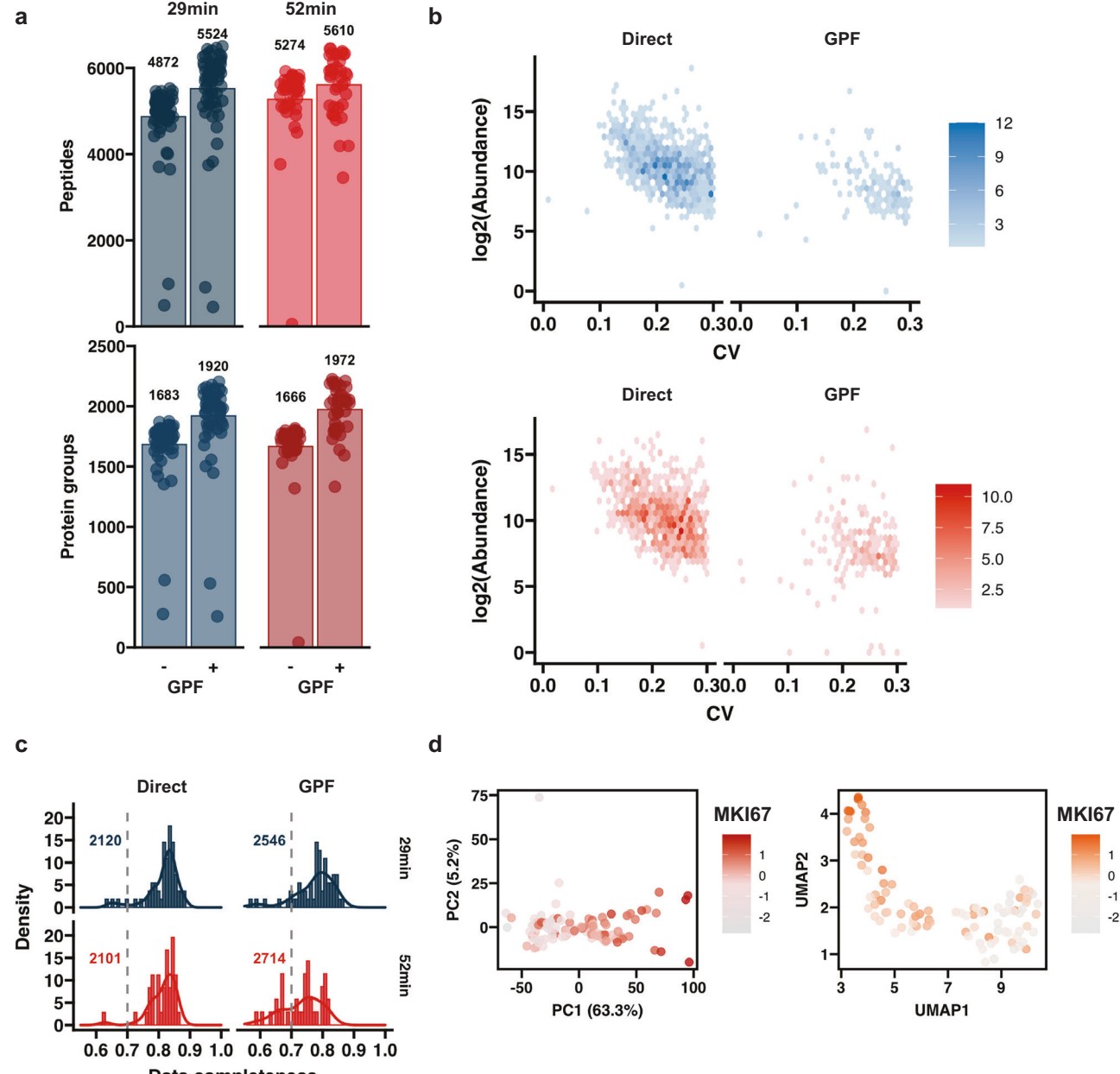

**Fig. 5 | Single-cell proteome profiling. a** Barplots of quantified peptides and proteins from single-cell inputs with two different methods. The spectra were searched either with directDIA approach or with a GPF-based library (GPF-DIA). **b** Hexbin plots showing the log2 transformed abundance and CV distribution for proteins quantified by directDIA and GPF. Twenty-nine minute in the top and 52 min in the bottom. **c** Histogram of data completeness for each cell. Dashed gray lines mark 70% complete detection. **d** Clustering of the integrated single-cell with PCA (left) and UMAP (right). Color coding denotes the standardized MKI67 protein abundance in each cell. Source data are provided as a Source Data file.

exhibiting <20% missing values with directDIA (Fig. 5c). However, GPF library application increased data sparsity to 30–40%. Arguably, this is an improvement for single-cell proteomics, as most studies to date have reported a high degree of missing values ~50%. In our case, with HEK293 being a rather homogeneous cell line, we expect that most of the variation in our data can be explained by differences in cell cycle stages. To further assess this, we integrated both the 40SPD and 24 SPD datasets by standardizing the abundances and clustered the cells with both linear (PCA) and non-linear (UMAP) methods to gauge this biological variation (Fig. 5d). The first principal component (PC1) captured a large degree or variation present in our dataset. To determine if PC1 was correlated with the cell cycle, we tracked the standardized abundance of the MKI67 protein, which has highest levels

during G2 and mitotic cell phases. There was a clear trend as the MKI67 levels increased along the PC1 (Fig. 5d). Similarly, in the UMAP analysis two clusters of cells were obtained and MKI67 levels increased along the second manifold dimension (Fig. 5d). No clustering based on run order was observed, however PC2 seemed to capture method related variation, but it should be noted the percentage of variation is rather small (Supplementary Fig. 7A, B), underlining that our workflow can capture biologically relevant trends in single-cell proteome profiles.

**scp-MS analysis of mouse embryonic stem cells reveals molecular and functional cell heterogeneity**
To further evaluate the ability of WISH-DIA to capture cellular heterogeneity, we carried out proteome profiling of mouse embryonic

stem cells (mESC) across two culture conditions[48–50]. We cultured cells in serum-free 2i condition (m2i) containing cytokine LIF with inhibitors of MEK and GSK3 pathways and in serum condition (m15) with cytokine LIF. The m2i cultured mESC state is referred as ground-state pluripotency, where cells express I pluripotency markers mimicking mouse epiblasts[48–50]. The serum-containing m15 conditions consist of a heterogeneous mix of undifferentiated and differentiating ESCs (Fig. 6a). To improve the throughput of our workflow to ensure sufficient cell numbers can be obtained in a timely manner, we adopted a faster LC/MS method capable of processing 72 cells per day (20 min run-to-run time). We processed and analyzed 599 cells, with >90% (548 cells) passing our quality control threshold (Supplementary Fig. 8A, see "Methods"). We noticed that we obtained around ~15% lower coverage for the m2i population compared to m15 (807 and 934 protein groups respectively)(Fig. 6b), however this is in line with the differences in size of these two cell types or reflect different culturing conditions (Supplementary Fig. 8B, D). Furthermore, the decreased overall number of proteins appears not to be a reflection of the chosen LC/MS method, but the nature of the chosen biological system. HEK293 cells analyzed with the same 72 SPD method showed similar coverage to Fig. 5 (Supplementary Fig. 5E), and therefore we attribute the lower proteome coverage to be a reflection of the lower proteome complexity in these primitive cell types when compared to HEK293.

To gauge the extent of cell heterogeneity present within the mESC populations, we used dimensionality reduction techniques (Fig. 6c). Both PCA and UMAP embeddings separated the m2i from m15 cells, with a tight m2i cluster and m15 subclusters, likely highlighting pluripotent and permissive states. DNA hypomethylation is a hallmark of m2i cells, while m15 cells have increased DNA methylation attributed to DNMT3A/B/L proteins[51,52]. Accordingly, we observed increased expression of the Dnmt3a protein in the m15 population compared to 2i (Fig. 6d). mESCs favor glycolysis over oxidative phosphorylation and bulk transcriptome analysis proposes an increased glycolytic preference of m15 cultures over 2i[50,52]. To investigate if there were systematic changes in these pathways we carried out gene-set enrichment analysis (GSEA)[53]. We could observe a clear preference for glycolysis over OxPhos for the embryonic-like population (Fig. 6e, f) and overall glycolysis was the most significantly enriched pathway (Supplementary Fig. 8G). Taken together, we conclude that our scp-MS approach is able to recapitulate known trends and could capture biological variation between the different media conditions and underlying cell states.

To gain deeper insight into which proteins are differentially expressed between the cell types we used a linear model approach to determine the most up- or down-regulated proteins (Fig. 6g). In line with the gene-set enrichment analysis, isocitrate dehydrogenase (Idh1) and glutamate dehydrogenase (Glud1) were among the top most significant proteins. These protein-level results mirror global trends across culture conditions, and highlight the increased glycolytic propensity for m15 cells relative to m2i cells[48,51,54]. The enzymes that provide donor molecules essential for demethylation (Idh1 and Glud1) and methylation (Mat2a) had contrasting expression profiles (Fig. 6h), which is interesting considering the pivotal role this modification plays in maintaining the embryonic stem cell state[55]. Furthermore, other enzymes involved in counteracting oxidative stress (Gsta4) and cholesterol synthesis (Fdps) were differentially expressed.

### Metabolic pathway regulation in embryonic stem cells

Given the differences in proteins involved in stem cell metabolism, we analyzed the proteins that govern the metabolites across glycolysis and oxidative phosphorylation in greater detail (Fig. 7a). By plotting the scaled abundance distribution of the embryonic and permissive stem cells we could clearly see that only select enzymes had altered protein levels (Fig. 7b). The ATP-dependent 6-phosphofructokinases (Pfkm, PfkI) and phosphoglycerate kinase 1 (Pgk1) remained stable, while the remaining quantified enzymes were upregulated in the

embryonic-like cells, albeit with notably different expression pattern. Based on the PCA and UMAP embedding, the m15 cells could be clustered into three subclusters (Fig. 7c). Accordingly, in all m15 cell clusters the Fructose-bisphosphate aldolase A (Aldoa) and alpha-enolase (Eno1) had decreased protein levels. In contrast, Glyceraldehyde-3-phosphate dehydrogenase (Gapdh) and Phosphoglycerate mutase 1 (Pgam1) had similar levels in the m15-1 subcluster similar to embryonic-like cells and was lower in m15-2 (Fig. 7d). This hints at heterogeneous glycolytic propensity of the identified m15 subcluster, potentially reflecting the extent to which cells have drifted from the embryonic-like state. Although the function of Eno1 in mESC has been recently explored[39], the exact role of Pgam1 has not yet been investigated.

Next, we evaluated the metabolic enzymes that are downstream of glycolysis (Fig. 7a). Again, we observed stable metabolic enzymes such as: Aconitase (Aco2), and pyruvate dehydrogenase complex subunits (Phda1, Phdb). However, the differential expression this time was bidirectional, as the enzymes were high in either embryonic-like or permissive cell populations (Fig. 7e). The Idh1 and Glud1 enzymes showed a peculiar schism, although both enzymes are responsible for generating alpha-ketoglutarate. Idh1, which generates the molecule from D-isocitrate is high in m2i cells, while Glud1, which performs the conversion from L-glutamate, is high in m15 (Fig. 7e). Idh1 has been extensively studied in the context of cancer and differentiation as it is tightly linked to TET function, which is essential in maintaining an stem cell state in healthy and malignant cells[55–58], underlining the biological significance of the quantified proteins. The cytoplasmic ATP-dependent citrate synthase (Acly) is more abundant in m2i cells, while the mitochondrial citrate synthase (Cs) remains stable. Overall, we demonstrate hypothesis generating potential of our WISH-DIA-based scp-MS workflow by tracking protein expression profiles for pivotal cellular processes.

## Discussion

In this study, we developed a label-free single-cell proteomics workflow by utilizing high sensitivity-tailored DIA methods in combination with latest chromatography and computational advances. Specifically, we show that DIA method design should be adjusted accordingly to sample load for optimal performance. We discovered that for low-input samples the detrimental dynamic range and chimeric spectra effects due to large isolation windows (>20 m/z) are overcome by increases in both resolution and injection time (Fig. 1). In contrast, the same trend was not observed for high-load. We adopt a DIA approach that solely relies on precursor-level quantification to further enhance sensitivity and use our findings to establish the WISH-DIA method. In tribrid instruments, the LIT can also be used to increase sensitivity while keeping isolation windows narrow[59]. We also applied the HRMS1 modification to LIT and showed that it significantly boosted proteome coverage for low-input samples (Supplementary Fig. 3). Finally, we showcase that WISH-DIA can be implemented on a range of chromatography platforms, consisting of both packed-bed and micropillar-array columns, with column- and gradient-specific data acquisition methods being required. As the latter are not compatible with EvoSep out-of-the-box, application of these columns at the time of writing requires alternative LC systems such as the Ultimate-3000 used in this work.

By applying WISH-DIA with micropillar-array-based chromatography we were able to achieve high proteome depth for low-input samples with appropriate sample throughput. We quantified ~5000 protein groups from 5–10 ng of input material, which is a highly relevant load for e.g. laser capture microdissection isolated tissue samples[60–62]. From ultra-low-input samples (250 pg) we manage to quantify >2000 protein groups which is often considered single-cell level input[2,3,12,18]. However, such inputs generated from bulk digest dilutions can be a poor proxy for true single-cell digests and numbers

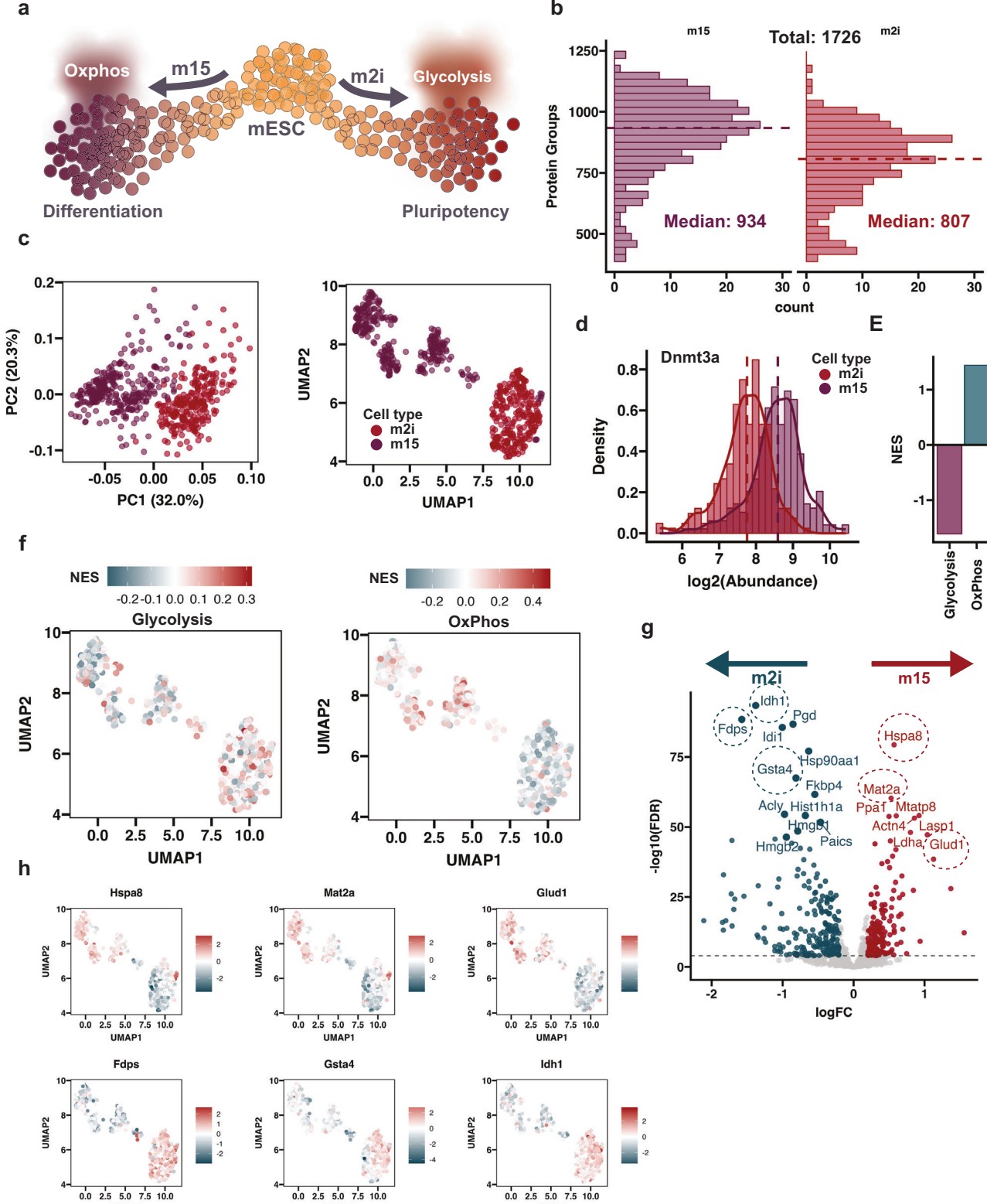

**Fig. 6 | Capturing biological variation in mouse embryonic stem cells.**
**a** Illustration of the used mESC model system. **b** Histogram showing the proteome coverage per cell, color denote the different cell types and dashed line the median number of protein groups, which is also annotated by text. The same coloring scheme is used for all subsequent figures. **c** PCA (left) and UMAP (right) embedding of the analyzed mESC cells. **d** Histogram with overlaid density line, showing log transformed expression of the Dnmt3a protein. **e** Curated GSEA results shown as a

barplot. **f** UMAP plots where the color gradient represents the normalized enrichment score (NES) for each cell for either glycolysis (left) or OxPhos (right). **g** Volcano plot showing top differentially expressed proteins, right side indicates proteins high in m2i and left in m15. Dashed circles mark proteins selected for further visualization. **h** Similar as **f**, but here the color gradient reflects the scaled protein abundance. Specific protein indicated above the plot. Source data are provided as a Source Data file.

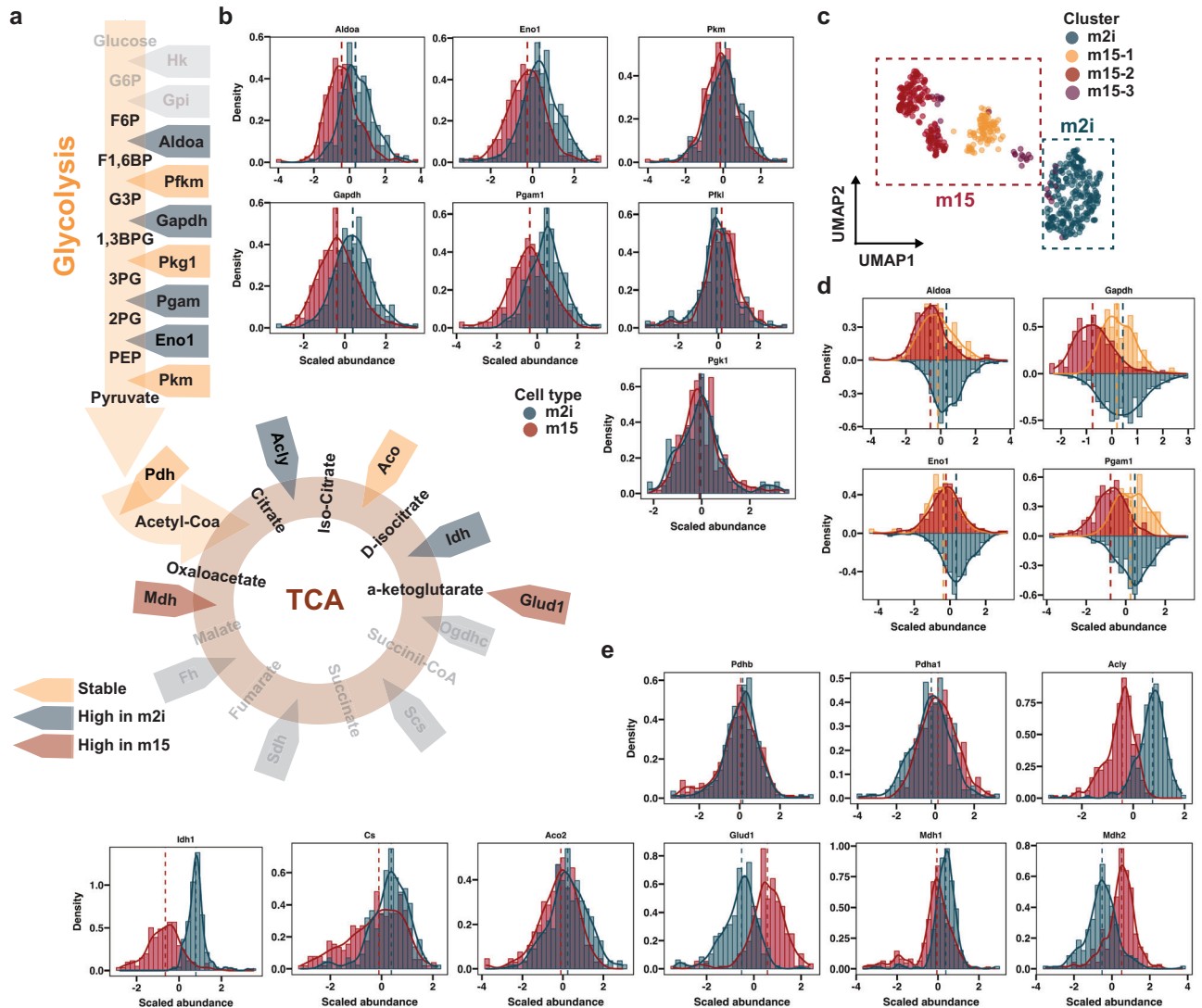

**Fig. 7 | Profiling of glycolytic and TCA enzyme expression at single-cell resolution. a** Simplified illustration of the simplified glycolytic and TCA metabolic pathways. Gray shading indicates proteins that were not quantified. **b** Histogram with overlaid density of scaled protein abundance. Dashed line indicates the median and specific proteins are visualized above the plot. **c** UMAP where the color indicates clusters obtained with gaussian mixture modeling (GMM). **d** Histogram with overlaid density of scaled, values from m2i, m15-1 and m15-2 are used. **e** Similar as b, but for proteins involved in TCA. Source data are provided as a Source Data file.

obtained with such samples should be interpreted with care. Accordingly, we tested our workflow with real single-cell digests and quantified ~2000 protein groups per single-cell at a throughput of 40 cells per day with the use of GPF libraries that boost the proteome coverage by >20% (Fig. 5a). Such libraries are a robust alternative to high pH for samples where offline fractionation is prohibitive, such as in the case of analyzing single cells. It should be noted that our entire workflow uses standardized lab equipment and does not require single-cell proteomics designated liquid handling systems as in other protocols[5,8,63], which should make the approach accessible for general proteomics labs and core facilities.

To accommodate the need for processing higher cell numbers in a biological context, we designed a method that can process 72 cells per day while maintaining reminiscent proteome coverage (Supplementary Fig. 5E). With this we profiled mESC cells that are either embryonic-like or are allowed to drift into differentiation permissible state (Fig. 7). We did not reach the proteome coverage that we saw in HEK293, but that is expected when less protein-rich cells are analyzed. The proteomic profiles recapitulated multiple known findings and presented how key metabolic enzyme expression is altered between

the different cell states. Interestingly, some of the identified enzymes, such as Idh1, Eno1 and Pgam1 are not only implicated in cell differentiation, but also malignant transformation[56,58,64,65]. This underlines the importance of the ability to monitor the expression of these key enzymes with single-cell resolution, as e.g. low-abundant cancer stem cell population might have a distinct expression profile that is obscured by more frequent cell types, when numerous cells are analyzed in cancer[1,66]. Studying the enzyme expression levels alone can provide valuable insights, however given the central role of metabolites in health and disease, being able to quantify these from the same cell should deliver unprecedented views of cellular states.

Although our label-free throughput is lower compared to DDA TMT-multiplexing based approaches, which can analyze up to 160 cells per day at a throughput of ~1000 protein groups per cell[4,7,10,11], the increased proteome depth and absences of a carrier channel and TMT quantification biases makes our LFQ workflow a solid and an easily implementable alternative. This might be of special relevance for patient samples where collecting sufficient cells for carrier samples might not be feasible. While we ran our experiments on an Orbitrap Eclipse Tribrid instrument, it is expected that WISH-DIA methods

translate directly to other Orbitrap platforms such as Exploris series instruments. Throughput can in principle be improved by adopting DIA compatible multiplexing, such as e.g. plexDIA, which has already been applied to single-cell analysis[6]. Other DIA compatible tags, such as Ac-IP or TMT complement ion quantification could be also explored to increase throughput[49,50]. Currently, our U3000-based workflow at 72 SPD would allow one thousand cells to be analyzed within two weeks, which is approaching a level of maturity capable of conducting biologically relevant interrogations of heterogeneous cell systems.

## Methods

### Cell culture and FACS sorting

HEK cells were cultured in RPMI media containing 10% FBS and 1% Penstrep. Upon reaching 80% confluence, cells were harvested and washed with ice-cold PBS to remove any remaining growth media prior FACS sorting and finally resuspended in ice-cold PBS at 1e6 cells/ml. E14 mESC (ATCC CRL-1821) were cultured on plastic plates coated with 0.1% gelatin (Sigma #G1393) in either "M15" media containing DMEM knockout (Gibco #10829), 15% FBS (Gibco #10270), 1xPen-Strep- Glutamine (Gibco #10378), 1xMEM (Gibco #11140), 1xB-ME (Gibco #21985) and 1000 U/ml Leukemia inhibitory factor (Merck #ESG1107) or in "2i" containing Ndiff 227 (Takara #Y40002), 3 μM CHIR99021 (Tocris #4423), 1 μM PD0325901 (Tocris #4192) and 1000 U/ml Leukemia inhibitory factor.

Cells were harvested the same way as for passaging. To distinguish between live and dead cells the harvested cells were washed with PBS then they were labeled with 0.1 μg/mL DAPI (4′,6-diamidino-2-phenylindole) (Invitrogen, Cat. No D1306) and were kept on ice until flow cytometry measurements.

Cell sorting for HEK293 cells was done on a FACS Aria III instrument, controlled by the DIVA software package (v.8.0.2) and operated with a 100 μm nozzle. For mESC C a Sony MA900 cell sorter using a 130 μm sorting chip was used. Cells were sorted at single-cell resolution, into a 384-well Eppendorf LoBind PCR plate (Eppendorf AG) containing 1 μL of lysis buffer (100 mM Triethylammonium bicarbonate (TEAB) pH 8.5, 20% (v/v) 2,2,2-Trifluoroethanol (TFE)). Directly after sorting, plates were briefly spun, snap-frozen on dry ice for 5 min and then heated at 95 °C in a PCR machine (Applied Biosystems Veriti 384-well) for an additional 5 min. Samples were then either subjected to further sample preparation or stored at −80 °C until further processing. All cell gating strategies are visualized in Supplementary Fig. 9.

HeLa cells (ATCC) were cultured in Dulbecco's Modified Eagle Medium (DMEM) for SILAC (Thermo Scientific, Cat#88364) that contains L-Glutamine, but neither L-Arginine nor L-Lysine, and supplemented with 10% dialyzed fetal bovine serum (Sigma-Aldrich, Cat#F0392) and 0.1% Penicillin/Streptomycin (Biowest, Cat#L0022). For stable isotope labeling, light and heavy media were prepared by adding 146 mg/L L-lysine and 84 mg/L L-arginine hydrochloride (light), and 152.8 mg/L L-lysine-13C6 and 87.2 mg/L L-arginine-13C6 hydrochloride (heavy) (Cambridge Isotope Labs, Andover, MA). Cells were cultured in 37 °C with 5% CO2 for 2 weeks (6 passages) to allow incorporation of stable isotopes before frozen. Freshly thawed cells were cultured in SILAC medium for two passages before harvest.

### Sample preparation of single cells for mass spectrometry

Single-cell protein lysates were digested with 2 ng of Trypsin (Sigma cat. Nr. T6567) supplied in 1 μL of digestion buffer (100 mM TEAB pH 8.5, 1:5000 (v/v) benzonase (Sigma cat. Nr. E1014)). The digestion was carried out overnight at 37 °C, and subsequently acidified by the addition of 1 μL 1% (v/v) trifluoroacetic acid (TFA). The resulting peptides were either directly submitted to mass-spectrometry analysis or stored at −80 °C until further processing. All liquid dispensing was done using an I-DOT One instrument (Dispendix).

### Liquid chromatography configuration

The Evosep one liquid chromatography system was used for DIA isolation window survey (Fig. 1) and HRMS1-DIA (Fig. 2) experiments. The standard 31 min or 58 min pre-defined Whisper gradients were used, where peptide elution is carried out with 100 nl/min flow rate. A 15 cm × 75 μm ID column (PepSep) with 1.9 μm C18 beads (Dr. Maisch, Germany) and a 10 μm ID silica electrospray emitter (PepSep) was used. Mobile phases A and B were 0.1% formic acid in water and 0.1% in Acetonitrile. The μPAC Neo limited samples column connected to the Ultimate 3000 RSLCnano system via built-in NanoViper fittings, and electrically grounded to the RSLCnano back-panel. For the single-column scheme the column was connected according to the "Ultimate 3000 RSLCnano Standard Application Guide" (page 38) and the autosampler injection valve, configured to perform direct injection of 1 μL volume sample plugs (1 μL sample loop−full loop injection mode). The pre-column scheme was also assembled according to the Standard Application Guide (page 47), a 20 μL injection loop was used. The analytical column was kept in a column oven and kept a constant temperature of 40 °C. The gradients used with the μPAC are as follows. Single-column scheme 20 min method: buffer B was increased from 1 to 12% (0–6.1 min), 12 to 17.5% (6.1–9 min), 17.5 to 35% (9–9.5 min), 35 to 99% (9.5–9.9 min), kept constant for 5 min (9.9 – 14.9 min) and dropped to 1% for 6 min (14.9–20 min). Single-column scheme 26 min method: buffer B was increased from 1 to 9% (0–6.1 min), 9 to 17.5% (6.1–11.5 min), 17.5 to 35% (11.5–13.7 min), 35 to 99% (13.7 –15.1 min), kept constant for 5 min (15.1–20 min) and dropped to 1% for 6 min (20–26 min). Single-column scheme 45 min method: buffer B was increased from 1 to 5% (0–6.1 min), 5 to 17.5% (6.1–26.5 min), 17.5 to 35% (26.5–32.7 min), 35 to 99% (32.7–33.1), kept constant for 6 min (33.1–39 min) and dropped to 1% for 6 minutes (39–45 min). Flow rate was kept at 250 nl/min from 6 to when the buffer B concentration was dropped to 1%. 500 nL/min used for the rest of the gradient. Pre-column scheme 29 min method: buffer B was increased from 1 to 7% (0–4.5 min), 4 to 20% (4.5–15 min), 20 to 40% (15–16.5 min) and 40 to 97.5% (16.5–21.5 min). Buffer B was then held constant for 5 min (21.5–26.5 min) and dropped to 1% and help constant for 3 min (26.5–29 min). Pre-column scheme 52 min method: buffer B was increased from 1 to 4% (0–4.5 min), 4% to 20% (4.5–26 min), 20 to 35% (26–37 min) and 40 to 97.5% (37–42 min). Buffer B was then held constant for 5 min (42–47 min) and dropped to 1% and help constant for 5 min (47–52 min). The flow rate was kept at 200 nL/min from 9 min to the points were the buffer B was dropped to 1%, 500 nL/min was used for the rest of the gradient (see Supplementary Fig. 4B). All the used Xcalibur methods are available in a repository. Both LC systems were coupled online to an orbitrap Eclipse Tribrid Mass Spectrometer (ThermoFisher Scientific) via an EasySpray ion source connected to a FAIMSPro device.

### MS data acquisition

The mass spectrometer was operated in positive mode with the FAIMSPro interface compensation voltage set to −45 V. Different DIA acquisition methods were used and are outlined in the results section or summarized in Supplementary Data 1 and 2. MS1 scans were carried out at 120,000 (except for HEK293 dataset collection where 240 K resolution was used) resolution with an automatic gain control (AGC) of 300% and maximum injection time set to auto. For the DIA isolation window survey a scan range of 500–900 was used and 400–1000 rest of the experiments. Higher energy collisional dissociation (HCD) was used for precursor fragmentation with a normalized collision energy (NCE) of 33% and MS2 scan AGC target was set to 1000%. For bulk peptide the samples were analyzed in triplicated (n = 3). For single-cell input for method development at least 5 cells (n ≥ 5) were measured per condition. For dataset collection n = 102 HEK293 cells and n = 599 mESC cells were analyzed.

## Data analysis

Spectronaut 16 and 17 versions were used to process raw data files. DirectDIA analysis was run on pipeline mode using modified BGS factory settings. Specifically, the imputation strategy was set to "None" and Quantity MS level was changed to MS1. Trypsin and Lys-C were selected as digestion enzymes and N-terminal protein acetylation and methionine oxidation were set as variable modifications. Carbamidomethylation of cysteines was set as fixed modification for experiments that used diluted Hela peptides and removed when single-cell runs were searched. The single-cell GPF library runs were added to directDIA to supplement the single-cell dataset search. SILAC experiments were processed in Spectronaut 16, with the Pulsar search engine setting altered to accommodate multiplexed samples. Two label channels were enabled and fixed Arg10 and Lys8 modifications were added to the second channel. The in-Silico Generate Missing channel setting was used with the workflow set to "label. The complete Spectronaut settings can be downloaded from the MassIVE repository (see "Data availability").

Protein and peptide quantification tables were then exported and analyzed in R or python (version 4.2.2) in the Visual Studio Code editor environment (version 1.73), with additional R packages: tidyverse[67], limma[68], and ggprism (https://csdaw.github.io/ggprism/). For python the following packages were used: numpy[69], pandas[70], scipy[71], UMAP[72], seaborn[73] and scikit-learn[74].

## mESC data analysis

The mESC raw data files were processed with Spectronaut 17 and protein abundance tables exported and analyzed further with python. First the proteome coverage and overall sample intensity was evaluated to remove poor quality cells from the dataset (Supplementary Fig. 8A). The proteome abundances were normalized sample-wise by subtracting the median of log transformed valalues and dividing by the median absolute deviation (robust z-transformation). The same operation was carried out protein wise, to remove any biases introduced by absolute protein abundance. Principal component analysis (PCA) was then carried out to identify global trends in the data. Cells that had a large distance in the first principal component were considered outliers and removed from further analysis (Supplementary Fig. 8F). The filtered data table was then exported and differential expression analysis was carried out with the use of the limma statistical package[68] in R. Gene-set nrichment analysis (GSEA) was carried out with the GSEApy[75] package in python with the MsigDB Hallmarks library.

Clustering of the mESC cells was carried out by using Gaussan-mixture modeling (GMM) with the scikit-learn package[74], where the number of clusters was set to 4 based on the qualitative characteristic of the PCA and UMAP (Fig. 7b). The final clustering presented in Fig. 7, was obtained by correcting UMAP clusters with the cluster annotation obtained from principal component values. The presented histogram of metabolic protein abundances were generated with the use of normalized protein values as described above. Overall, all basic analysis was carried out in python and R was predominantly used for data visualization, except for the case of differential expression. For analysis code and tables see "Data availability".

## Hela tryptic digest preparation

Cells were harvested at 80% confluence and lysed in 5% sodium dodecyl sulfate (SDS), 50 mM Tris (pH 8), 75 mM NaCl, and protease inhibitors (Roche, Basel, Switzerland, Complete-mini EDTA-free). The cell lysate was sonicated for $2 \times 30$ s and then was incubated for 10 min on ice. Proteins were reduced and alkylated with 5 mM tris(2-carboxyethyl)phosphine (TCEP) and 10 mM CAA for 20 min at 45 °C. Proteins were diluted to 1%SDS and digested with MS grade trypsin protease and Lys-C protease (Pierce, Thermo Fisher Scientific)

overnight at an estimated 1:100 enzyme to substrate ratio quenching with 1% trifluoroacetic acid (TFA) in isopropyl alcohol. For the cleanup step by styrenedivinylbenzene reverse-phase sulfonate (SDB-RPS)[76], 10 µg of peptides was loaded on StageTip[43] and washed twice by adding 100 µL of 1% TFA in isopropyl alcohol. Peptides were eluted by adding 50 µL of an elution buffer (1% Ammonia, 19% ddH2O,and 80% Acetonitrile) in a polymerase chain reaction (PCR) tube and dried at 45 °C in a SpeedVac. Lastly, peptides were resuspended in buffer A and their concentration was measured by nanodrop.

## Reporting summary

Further information on research design is available in the Nature Portfolio Reporting Summary linked to this article.

## Data availability

The complete MS raw data, Spectronaut search files have been deposited to MassIVE under the following accession MSV000090792 (https://doi.org/10.25345/C5JM23M36). mESC raw data were deposited into a separate repository with the following accession MSV000092429 (https://doi.org/10.25345/C5DB7W12H). The processed data used to generate the figures can be accessed from two Zenodo repositories: https://doi.org/10.5281/zenodo.7433298v and https://doi.org/10.5281/zenodo.8146605. The specific link between the tables together with the code required to recreate the figures is stored in a separate repository (see "Code availability"). The comparison single-cell data was downloaded from the following PRIDE repository: PXD024043. MsigDB Hallmarks library was accessed via the GSEApy[75] package. Source data are provided with this paper.

## Code availability

The code used to generate to process the tables exported from Spectronaut analysis has been stored in the following repository: https://github.com/Schoof-Lab/WISH-DIA. The required tables for the code are provided in Zenodo repositories: https://doi.org/10.5281/zenodo.7433298 and https://doi.org/10.5281/zenodo.8146605 An archived version of the repository can be accesses here: https://zenodo.org/badge/latestdoi/577804073.

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

## Acknowledgements

We would like to thank Robert van Ling at ThermoFisher for early access to the µPAC Neo Low Load column and pre-column, and EvoSep for elaborate collaborations on the EvoSep One instrument. We also thank Biognosys for pre-release access to Spectronaut 17. Some of this work was funded by a grant from the Novo Nordisk Foundation to E.M.S. with reference number NNF21OC0071016. B.F. is the recipient of a fellowship from the Novo Nordisk Foundation as part of the Copenhagen Bioscience PhD. Programme, supported through grant NNF19SA0035442. V.P. is funded by a Leo Foundation grant awarded to S.F.T and E.M.S. (LF-OC-21-000832). P.A.F. is funded by a Danish Cancer Society grant (R324-A17978). Work in the B.T.P. lab is supported by grants from the Svend Andersen Foundation, the Candys foundation, the Danish Cancer Society, Independent Research Fund Denmark and through a center grant from the Novo Nordisk Foundation (Novo Nordisk Foundation Center for Stem Cell Biology, DanStem; Grant Number NNF17CC0027852). U.A.D.K. acknowledges funding by a Novo Nordisk Foundation Young Investigator Award (NNF16OC0020670). The S.G. group is supported by Novo Nordisk Foundation Grants NNF19SA0056783, NNF20SA0066621, and NNF19SA0057794. We also thank all members from the Cell Diversity Lab, headed by E.M.S. for constructive input and fruitful discussions, and the DTU Proteomics Core for technical instrument support.

## Author contributions

E.M.S. and V.P. conceived and designed the project. V.P., N.U., P.A.F., S.L.S., G.K., T.P and E.M.S. performed experiments, and B.F., J.O.D.B., S.F.T., U.A.D.K., B.T.P., S.G. and K.N.N. provided critical input. Data analysis was performed by V.P. The manuscript was drafted and revised by V.P. and E.M.S., with input from all other authors. E.M.S. supervised the work.

## Competing interests

J.O.D.B. is an employee at Thermo Fisher Scientific. All other authors declare no competing interests.

## Additional information

[1]Department of Biotechnology and Biomedicine, Technical University of Denmark, Søltofts Plads 224 2800 Kgs, Lyngby, Denmark. [2]The Finsen Laboratory, Rigshospitalet, Faculty of Health Sciences, University of Copenhagen, Copenhagen, Denmark. [3]Biotech Research and Innovation Centre (BRIC), University of Copenhagen, Copenhagen, Denmark. [4]The Novo Nordisk Foundation Center for Protein Research, Faculty of Health Sciences, University of Copenhagen, Copenhagen 2200, Denmark. [5]Department of Proteomics and Signal Transduction, Max-Planck Institute of Biochemistry, Martinsried 82152, Germany. [6]MaxPlanck Institute of Biochemistry, Martinsried 82152, Germany. [7]Thermo Fisher Scientific, Technologiepark-Zwijnaarde 82, B-9052 Gent, Belgium. [8]Department of Dermatology, Bispebjerg Hospital and Department of Biomedical Sciences, University of Copenhagen, Copenhagen, Denmark. [9]Dept of Clinical Medicine, University of Copenhagen, Copenhagen, Denmark. ✉e-mail: erws@dtu.dk

