## [Peer Review File · Nature Communications]

Reviewers' Comments:

Reviewer #1:

Remarks to the Author:

Thank you for the opportunity to read and review "Enhancing single-cell proteomics through tailored Data Independent Acquisition and micropillar array-based chromatography"
As there is considerable debate in the final form that single cell proteomics by mass spectrometry will need to take to become a biological tool, this work is both timely and important.
The observation that coisolation interference effects decrease at single cell loads is counter-intuitive but well-supported.

My one major comment is that these approaches do not appear to be compared to recently published work by Andreas-David Brunner. The work is included in the citations, but a comparison between these two approaches might be warranted as this appears to largely be a method comparison paper? However, I could also be convinced against this being a real requirement.

Otherwise, I only have minor comments for this work and consider it an important chapter in the early evolution of single cell proteomics and suitable for this journal.

Minor comments:

- 1) While work with WISH-DIA was largely based on the HRMS1 study by Xuan et al., that work was likewise based on the pSMART and BoxCar work from Prakash et al., 2014 and Meier et al., 2018, respectively. I think the value of these studies in optimizing cycle time for deep sample analysis warrants at least a citation in addition to the original HRMS1 work.
- 2) In the massive repository (MSV000090792) a "READ ME" file is indicated as present, but I was not able to locate this. Please check the repository to verify that this MetaData is provided for the study. The repository is well organized and the files are well-named, but a central document would be helpful.
- 3) Figure S6: Please include a color key. I found the grouping a little hard to follow without one.

Reviewer #2:

Remarks to the Author:

The authors performed a systematic optimisation of a DIA method for low input material. They surveyed different window sizes and resolutions for different input materials and reported superior performance of large windows for low input material. They then combine large window DIA with HRMS1 and MS1 quantification to further increase ITs. Beside directDIA they also tested library based DIA with libraries generated with GPF or with increased sample material ("high-load" library). They applied the method to measure HEK293 single cells and captured cell cycle dependent changes of MKI67.

The authors do not introduce any new concepts but rather optimize a DIA scheme and combine it with existing strategies (HRMS1). The advantage of large windows is not surprising but the reported results demonstrate the power of label-free DIA for single cell applications. Despite the lack of novelty, the work could be interesting for labs that want to set up DIA methods for single cell/low input analysis. The biological application is a bit disappointing and the authors need to expand on this to show the general utility of the method.

Further points that need to be addressed:

-I was surprised to see that the authors changed the method for the single cell application (line 386) and didn't use the method they optimised. The authors should have made a more systematic optimisation i.e. include lower sample amounts in their optimisation. Especially the window size survey was only down to 1ng which is not sufficient if they want to apply it to single HEK cells.

-Also the measurements were conducted on 2 different chromatographic systems (evosep one and ultimate 3000). Could the authors comment why they have done this. The performance of the developed DIA scheme is dependent on the chromatography and should be ideally optimized on one

system and with one column. Further the ultimate was operated at a different flow rate- how did this impact the sensitivity of the proposed method?

-Also optimum window size depends on gradient length as this also has an impact on peak width and consequently at the number of points per peak. For example which method used for Figure 1B (31min or 58min). The authors should at least discuss this aspect.

-The authors should compare their method to existing methods applied on low input material (e.g. download from literature). Despite the difficulties and limitations of comparing methods I think it would be important for the reader to put the reported numbers into context.

-Figure 1C and 2B: Could the authors comment on the 2 distributions in the plots and can they exclude any artifacts from the estimation of the datapoints per peak

-Line 240 Wrong Figure referenced

-Line 242: "The additional points are detected potentially due to longer ITs which allows quantification of the elution profile tails that fall below the background intensity at shorter ITs." This is very speculative. Could the authors show examples? How exactly are points / peak calculated?

-Line 236: "suggesting that the chimeric spectra effects due to co-isolation at such loads are sufficiently low." I would suggest changing the wording here. The problem of co-isolation is probably not diminished at lower loads but rather outweighed by the loss of identified precursors due to limited sensitivity.

-Line 276" The extra identifications by HRMS1 primarily arose from low-abundant proteins (Figure S2B)." Figure S2B needs more explanation. It is very surprising that there is no trend in identifications based on abundance. This is normally not the case for DIA methods- can the authors give an explanation?

-Figure 2D: indicate injection amount in legend

-Figure 2F: maybe label the ratios H/L instead of L/H as this corresponds to the x-axis label? Ie. 5:1 instead of 1:5

-Also Figure 2F: Can the authors explain why the 1:1 ratio is off? This is rather unexpected. Can the authors exclude pipetting errors?

-Line 296: "There was a clear drop in accuracy as the ratio of light and heavy peptides was increasing." I assume the authors ment "ratio of heavy and light"

-Also 296: "There was a clear drop in accuracy as the ratio of light and heavy peptides was increasing, potentially, due to the decreasing proportion of light peptides in the samples making them harder to quantify." This conclusion sounds oversimplified. Is the reason a non-linear response of peptides and how do the response curves differ in MS1 and MS2? - the author could plot response curves for some example peptides. Also could the authors estimate the impact of interferences on MS1 and MS2?

-In general the authors make the accuracy benchmarks based on visually comparing the distributions (e.g. Figure 2F). Could the authors instead do a more quantitative comparison (e.g. comparing offsets of ratios and/ or distribution widths)

-334 "Retention times were very robust across runs, with almost all precursor elution apex deviations being limited to 2.5 seconds (Figure 3E), underlining the solid chromatographic performance of this novel uPAC Neo Low Load column" Could the authors put this into context. E.g. putting a quantitative value on the variation and comparing it to other columns/technologies? (e.g. data from literature or in-house?)

-Figure 3A: This Figure is out of context and I would remove it. I would also recommend to remove "micropillar array-based chromatography" from the title as the authors did not do much

optimisation/benchmark based on their chromatography (see comment above)

-Figure 3D: could the authors comment on the 2 peaks in Figure 3D. Is that related with how the FWHM is calculated. Could the authors give more details on how such parameters have been calculated and could cause this.

-Figure S4 B legend: "Barplots showing numbers of proteins with single-column and pre-column configuration." Could the authors label the Figures accordingly so that it is clear which numbers are related to pre-column and which to single column. Also it says Spectronaut16 "pre-release" instead of Spectronaut17 in the legend.

-Figure 4 legend: seems wrong e.g. B "The same as A, but with the uPAC pre-column in-line" does not correspond to Figure 4B. Also C, D and E labels seem not to match the corresponding figure

-Line 434: "This was primarily driven by the addition of a large number proteins with low quantitative accuracy (Figure 5B)." I do not understand the relation to Figure 5B here. This needs to be better explained.

-Line 435: "The quantitative accuracy of the proteins that could be identified by directDIA remained similar, however some detrimental effects due to the library search can be noted, likely due to the detection of additional low-abundant peptides (Figure S5)." This benchmark is very qualitative and the "detrimental effect" needs more explanation as this is not clear from Figure S5.

-Line 417: "However, the exact impact of using.. are potentially lost during single-cell processing might be erroneously quantified." This statement would indicate that the FDR control is wrong- could the author comment on this or modify the statement accordingly.

-Figure 6B: QBL is mentioned in the legend but not explained. Figure needs more explanation in legend and/or text. In general I would recommend expanding Figure legends as some Figure legends have very little information.

-Line 486: "...(PCA) and non-linear (UMAP) methods to gauge this biological variation (Figure 5D)." This should be Figure 6D? Same in line 492

Reviewer #3:

Remarks to the Author:

Petrosius et al. describes an analysis of orbitrap-based data-independent acquisition (DIA) for ultra-low-input samples. They describe a difference in optimal yield conditions between high- and low-input samples, and go on to recommend a protocol using standardized lab equipment that they claim is optimal for single-cell proteomics. The presented techniques are not novel, but the optimization is. The presented protocol seems timely and useful. I have but two minor comments:

* I found it hard to follow the difference between HRMS1 and WISH-DIA. It seems like HRMS1 is a good enough term for the protocol, as it is just a special case.

* It is hard to spot the differences in the presented violin plots. It would be easier to see reported differences in e.g. Figure 2F, and Figure A&B if scatterplots were used.

We would like to thank all the reviewers for evaluating our work and welcome their expressed concerns, which we hope to have addressed and can only agree have made the manuscript stronger. We especially aimed to address the two major concerns raised by the reviewers (comparing to other workflows in the field and showcasing biological application) and adjusted the manuscript accordingly. Specifically, we added a supplemental figure comparing our single-cell results to Brunner et al. 2020, which is the main label-free DIA based single-cell proteomics paper published to date. We re-searched a representative portion of their raw data with the same software as ours (Spectronaut), using only directDIA (i.e. without the use of spectral libraries) and find that their results for the cell population in a similar cell cycle stage as ours are comparable, and that our data even seems to slightly outperform their workflow. We also took the opportunity to conduct a pilot study with biological implication, to showcase the ability of WISH-DIA to discern cell-types and detect cell heterogeneity in a complex cell differentiation model. Briefly, we used a mouse embryonic stem cell (mESC) system and studied cellular heterogeneity between embryonic and permissive stem cell states, revealing major differences in the context of metabolic enzymes involved in glycolysis in TCA.

To ease the navigation of the rebuttal we have color coded the text where:

- Black - comment from reviewer
- Blue - our response
- Green - text added to the manuscript (with indicates lines)

REVIEWER COMMENTS

Reviewer #1 (Remarks to the Author):

Thank you for the opportunity to read and review "Enhancing single-cell proteomics through tailored Data Independent Acquisition and micropillar array-based chromatography"

As there is considerable debate in the final form that single cell proteomics by mass spectrometry will need to take to become a biological tool, this work is both timely and important.

The observation that coisolation interference effects decrease at single cell loads is counter-intuitive but well-supported.

My one major comment is that these approaches do not appear to be compared to recently published work by Andreas-David Brunner. The work is included in the citations, but a comparison between these two approaches might be warranted as this appears to largely be a method comparison paper? However, I could also be convinced against this being a real requirement.

Otherwise, I only have minor comments for this work and consider it an important chapter in the early evolution of single cell proteomics and suitable for this journal.

We thank the reviewer for their comment that this work is timely and important. We also agree with them that a comparison to the Brunner et al. data is warranted, which we aimed to accomplish

now in the revised manuscript. To directly evaluate and compare the performance of both studies, we needed to reanalyse the raw data from Brunner et al., as the results originally shown were based on a 50ng spectral library and DIA-NN. Using Spectronaut, we re-analyzed 10 randomly selected samples from Brunner et al. As we used an asynchronous population, we chose RAW files from the repository labeled “UB”, which based on the method section should correspond to a non-cell cycle stage enriched sample. The results suggest our results to be highly comparable, even when using our faster 72SPD (72 samples per day) method we developed for the revised manuscript using the U3000 combined with uPAC Neo. The figure with the identified protein count has been added to the supplementary Figure S5E.

Rebuttal Figure 1. Bar of detected number of proteins (right) and peptides(left) from our study and Brunner et al, 2022. The protein bar plot is incorporated as the supplemental figure S5E.

Minor comments:

1) While work with WISH-DIA was largely based on the HRMS1 study by Xuan et al., that work was likewise based on the pSMART and BoxCar work from Prakash et al., 2014 and Meier et al., 2018, respectively. I think the value of these studies in optimizing cycle time for deep sample analysis warrants at least a citation in addition to the original HRMS1 work.

We completely agree that these references should be added, thank you for pointing them out. They have been included together with Xuan et al in the main text.

2) In the massive repository (MSV000090792) a "READ ME" file is indicated as present, but I was not able to locate this. Please check the repository to verify that this MetaData is provided for the study. The repository is well organized and the files are well-named, but a central document would be helpful.

We have double checked the presence of the mentioned “README” file that links the figures in the manuscript to the raw data and it appears to be present. We are confused as to why it might not have been visible to the reviewer at the time of review. In case the issue persists, here is a

[link that should allow direct download from massIVE \(verified on multiple computers locally\) f.MSV000090792/updates/2022-11-29_Valdemaras_dff243a6/metadata/README.txt](https://f.MSV000090792/updates/2022-11-29_Valdemaras_dff243a6/metadata/README.txt) .

3) Figure S6: Please include a color key. I found the grouping a little hard to follow without one.

Color key has been added, thanks for this suggestion

Reviewer #2 (Remarks to the Author):

The authors performed a systematic optimisation of a DIA method for low input material. They surveyed different window sizes and resolutions for different input materials and reported superior performance of large windows for low input material. They then combine large window DIA with HRMS1 and MS1 quantification to further increase ITs. Beside directDIA they also tested library based DIA with libraries generated with GPF or with increased sample material (“high-load” library). They applied the method to measure HEK293 single cells and captured cell cycle dependent changes of MKI67.

The authors do not introduce any new concepts but rather optimize a DIA scheme and combine it with existing strategies (HRMS1). The advantage of large windows is not surprising but the reported results demonstrate the power of label-free DIA for single cell applications. Despite the lack of novelty, the work could be interesting for labs that want to set up DIA methods for single cell/low input analysis. The biological application is a bit disappointing and the authors need to expand on this to show the general utility of the method.

We thank the reviewer for their evaluation of our work, and their recognition of key developments towards better single-cell proteomics workflows it encompasses. We agree that the biological application of the original manuscript was rather limited, and have tried to address this in the revised manuscript by applying it to a stem cell differentiation system, adding completely new Figures 6 & 7. Briefly, we profiled mESC cells that are either in an embryonic-like, or more permissive state, similar to the original cell system published in Kolodziejczyk et al., Cell Stem Cell 2015. The proteomic profiles measured by WISH-DIA recapitulated multiple known findings and presented how key metabolic enzyme expression is altered between the different cell states. Results are mainly described in the manuscript from LINES 548 to 649.

Further points that need to be addressed:

-I was surprised to see that the authors changed the method for the single cell application (line 386) and didn't use the method they optimized. The authors should have made a more systematic optimisation i.e. include lower sample amounts in their optimisation. Especially the window size survey was only down to 1ng which is not sufficient if they want to apply it to single HEK cells.

We apologize for the lack of clarity in this regard, which we tried to address in the revised manuscript. In our original window size optimization we only went down to 1ng, however for the uPAC column we carried out similar experiments using 10ng all the way down to 250pg input. We did not incorporate this into the original manuscript, to avoid reiterating the identical finding with just a different analytical column. To amend this, we have included a supplementary table, peptide and protein groups counts are noted. This is included in the manuscript now as Supplemental Table 1, and addressed by the following line:

Line 404-407 “Optimal methods were identified for each gradient length by carrying out similar isolation window experiments as previously described (Figure 1-2, Table S2) and the best performing methods for all configurations and inputs are summarized in Figure S5A.”

For true single-cell, increasing the injection time further to 502 ms (240K resolution) increases the proteome coverage (Figure S4C), while it does not have the same effect on 250pg samples, underlining the pitfall of assuming that such type of sample is comparable to single-cell. We hope with the included table, and Supp. Fig. 5A and Table S2, it will be clear how the optimal methods for single-cell measurements were reached.

Figure S5C. Barplot showing the number of quantified peptides and protein groups with the 120k and 240k WISH-DIA methods.

-Also the measurements were conducted on 2 different chromatographic systems (evosep one and ultimate 3000). Could the authors comment why they have done this. The performance of the developed DIA scheme is dependent on the chromatography and should be ideally optimized on one system and with one column. Further the ultimate was operated at a different flow rate- how did this impact the sensitivity of the proposed method?

We are happy to clarify the experimental design, and apologize the lack of clarity in this regard in the original manuscript. We have now added a brief discussion to the revised manuscript in this regard. We started out with the EvoSep platform as a proof-of-concept, to evaluate and demonstrate that optimal Orbitrap parameters (i.e. resolution and window size) for DIA are

dependent on the injected sample amount. To validate these observations on an independent chromatography platform, we switched to the U3000 system combined with the uPAC Neo Low Load columns to serve as an independent validation. In both cases, specific WISH-DIA methods were optimized according to the specific chromatography platform characteristics (e.g. peak width and gradient lengths) to maintain acceptable points-per-peak. In addition, with their superior separation power, we assumed that the uPAC columns would be more suitable for single-cell proteomics. At the moment of writing, the uPAC Neo low load column is incompatible with current standardized EvoSep methods, hence the requirement for an alternative LC platform to operate this column. According to the manufacturers, this is due to the non-porous nature (and consequently, low retentiveness) of the Low Load version of the uPAC, compared to the standard PepSep columns. Furthermore, for the Evosep platform the sample needs to be transferred to the Evotip, which we feared would introduce significant material losses (e.g. FIG S5D), resulting in decreased proteome coverage and sample variations introduced by additional pipetting steps. The U3000 platform allows us to perform direct injection from the 384w plate, circumventing this costly step. The U3000 was operated at standard nano-flowrates of 200-250nl/min (pre-columns and single-column configuration), which in theory should result in slightly lower sensitivity than the 100nl/min of EvoSep Whisper methods, but appears to be compensated for by the sharper peaks of the uPAC Low Load column. We agree it would be interesting to follow up on decreasing flowrates on the uPAC for a future manuscript, but were unsuccessful on the U3000 and would likely require a Vanquish Neo.

Figure S5D. Barplot showing the number of quantified peptides from single-cell input with pre-column configuration

Lines: 371-373 “Next, we substituted the packed C18-beads column with a next-generation uPAC Neo Low Load column to further augment our low-input workflow efforts (Figure S4A) and explore the general applicability of WISH-DIA schemes across different chromatography platforms.”

Lines: 662-666 “Finally, we showcase that WISH-DIA can be implemented on a range of chromatography platforms, consisting of both packed-bed and micropillar array columns, with column- and gradient-specific data acquisition methods being required. As the latter are not

compatible with EvoSep out-of-the-box, application of these columns at the time of writing requires alternative LC systems such as the Ultimate-3000 used in this work. “

-Also optimum window size depends on gradient length as this also has an impact on peak width and consequently at the number of points per peak. For example which method used for Figure 1B (31min or 58min). The authors should at least discuss this aspect.

We completely agree with the reviewer on this. As the majority of the optimization was carried out at 40SPD (31min) method and only a single method (Figure 2A-C) used the 20SPD method, we initially refrained from commenting on this aspect to try and not overload the reader with too many variables at once. We have now added a small description to indicate the need to accommodate gradient length in one's method design, and clarify the column-specific parameters (i.e. EvoSep and uPAC separately) in Figure 1-2 and Table S1. We hope this further exemplifies the considerations to be made when designing such methods, and that our window size evaluations were conducted on the same gradient lengths to showcase only the impact of window size versus sample load, and not chromatographic peak widths.

Lines: 276-280 “The scan-cycle has to be coordinated with the chosen chromatographic method to ensure that enough data points per elution peak are acquired to maintain robust sampling³¹. Varying the active gradient length can affect the peptide elution peak width and the chosen scan-cycle time should be coordinated with this timeframe³². With our chosen parameters...”

-The authors should compare their method to existing methods applied on low input material (e.g. download from literature). Despite the difficulties and limitations of comparing methods I think it would be important for the reader to put the reported numbers into context.

We thank the reviewer for this suggestion, which we agree would help the reader evaluate the performance of our workflow in comparison to previous results. However, a lot of the low-input material DIA results in the public domain are confounded by alternative search strategies (e.g. library search, co-searching with high-load samples, etc) and a wide range of search engines (e.g. MSFragger, DIA-NN, Spectronaut). To simplify such a comparison, and given that the primary focus of WISH-DIA is towards single-cell proteomics, we now added a comparison on true single-cell input from Brunner et al, 2022 (also see response to reviewer #1- Rebuttal figure 1), which we hope the reviewer agrees is the most important comparison. Here we see that we are performing comparably if not slightly better to the recent state-of-the-art LFQ DIA results (more details provided in response to reviewer #1).

“Accordingly, direct injection boosted our average identifications by ~60% for the shorter and ~30% for the longer method (Figure 3C), bringing our quantified protein numbers to 1151 and

1318 when searched with directDIA, which is highly comparable to coverage obtained with low-input specialized instruments (Figure S5E).”

-Figure 1C and 2B: Could the authors comment on the 2 distributions in the plots and can they exclude any artifacts from the estimation of the datapoints per peak

We thank the reviewer for requesting more details on this interesting observation. In the standard, pre-defined Evosep+PepSep column set up, there are two clearly notable peptide elution types that are dependent on the retention time and intensity, with the latter potentially being more dominant. One can clearly see a sharpening of elution peaks as the end of the active gradient is reached, resulting in a distinct bimodal distribution of peak widths, which appears to be highly retention time dependent.

Rebuttal Figure 1. 2D density distribution of precursors peak full-width half maximum and elution apex time in the DIA window optimization survey for 10ng input

To ensure that this reflected the actual raw data, we also looked into the chromatograms that showed peak sharpening at the end of the chromatographic gradient. This might be caused by the flow ramping at the end of the Evosep gradient, and is unfortunately out of the user's control.

Rebuttal Figure 2. Chromatogram of a selected 10ng run visualized in FreeStyle software. 4 peaks are selected below the TIC chromatogram to show the clear width difference between the start and end of the active gradient.

-Line 240 Wrong Figure referenced

Thank you, this has now been fixed.

-Line 242: “The additional points are detected potentially due to longer ITs which allows quantification of the elution profile tails that fall below the background intensity at shorter ITs.” This is very speculative. Could the authors show examples? How exactly are points / peak calculated?

The points per peak are taken from the Spectronaut software itself. Spectronaut reports the number of scans for a precursor or fragment within the start and end of elution as the data points per peak. We assumed that the longer injection times facilitate a signal for a precursors that otherwise is too low in copy number to produce a signal at the tails of the elution peak. To provide some evidence for your hypothesis, we selected a few specific precursors that showed widening in their peak width as injection time was increased. In the selected example cases, that show distinct trends. One where the peak width increase rather rapidly from 10mz to 20mz method and one where it increases linearly. Overall, there is notable increase in the peptide elution peak width and accordingly the obtained data points per peak (Rebuttal Figure 5). Since the major variable between the different data points is the chosen MS method (injection time/resolution and isolation window), it is likely that the observed trend is dependent on this. However, to ensure that it is not just an artifact produced by the software we, we also check at the most intense transition that is used for quantification for these two peptides in the raw data with the use of Freestyle. Although for the first peptide the peak shape is not ideal, one can mark the difference in peak shape and width as the injection time is increased (Rebuttal Figure 6). For the second peptide, the peak shape is nicely gaussian and we hope the reviewer can appreciate the overall increasing intensity of the peak, that in turn makes the tails of the distribution also more intense allowing them to be incorporated as points per peak for quantification (Rebuttal Figure 7). This trend is observed for some, but not all peptides, which is potentially affected by AGC control.

Rebuttal Figure 3. Scatter plots with a line connecting the points by replicate. Full width at half maximum (FWHM) is plotted on the y-axis and the method isolation window on the x-axis. The specific peptide sequence is noted above the plot.

Rebuttal Figure 4. Chromatograms for the GEMMDLQHGSLFLQTPK peptide. The TIC based chromatogram is visualized in the top, followed by 80mz, 40mz, 20mz and 10mz DIA methods.

Rebuttal Figure 5. Chromatograms for the ESYSIYVYK peptide. The TIC based chromatogram is visualized in the top, followed by 80mz, 40mz, 20mz and 10mz DIA methods.

-Line 236: “suggesting that the chimeric spectra effects due to co-isolation at such loads are sufficiently low.” I would suggest changing the wording here. The problem of co-isolation is probably not diminished at lower loads but rather outweighed by the loss of identified precursors due to limited sensitivity.

Thank you for this excellent suggestion. The text has been modified accordingly.

Lines: 275-276 “...suggesting that the chimeric spectra effects due to co-isolation at such loads are outweighed by increased resolution and IT that enhance the sensitivity.”

-Line 276” The extra identifications by HRMS1 primarily arose from low-abundant proteins (Figure S2B).” Figure S2B needs more explanation. It is very surprising that there is no trend in identifications based on abundance. This is normally not the case for DIA methods- can the authors give an explanation?

We apologize for the lack of clarity in our figure. There is a trend in identification based on abundance. To avoid any confusion, we have remade this figure as a histogram, where the detected proteins with either standard DIA (OT-DIA) or HRMS1 (HRMS1-DIA) are binned based on their abundance. We hope that one can clearly see now that HRMS1 detects a similar number of proteins in the upper abundance range, but a substantially higher number in the lower range. This figure has now been replaced and made into Figure S2B.

Figure S2B. Histogram showing detected number of proteins at a certain log transformed abundance bin with HRMS1 or standard DIA.

-Figure 2D: indicate injection amount in legend

The injection amount has now been added, thanks.

-Figure 2F: maybe label the ratios H/L instead of L/H as this corresponds to the x-axis label? i.e. 5:1 instead of 1:5

Thank you for the helpful suggestion, we corrected the axis label.

-Also Figure 2F: Can the authors explain why the 1:1 ratio is off? This is rather unexpected. Can the authors exclude pipetting errors?

The same pool of mix was used for both standard DIA and HRMS1 DIA. As we only see these biases in one of the methods, we would speculate that the bias arises due to inaccurate deconvolution of the heavy and light peptide signal (potentially due to co-elution on the MS1 level). We do not see this bias in MS2 based quantification, suggesting the overall dilution mixes to be correct.

-Line 296: "There was a clear drop in accuracy as the ratio of light and heavy peptides was increasing." I assume the authors ment "ratio of heavy and light"

Thank you for pointing out our oversight. The main text has been fixed.

-Also 296: "There was a clear drop in accuracy as the ratio of light and heavy peptides was increasing, potentially, due to the decreasing proportion of light peptides in the samples making them harder to quantify." This conclusion sounds oversimplified. Is the reason a non-linear response of peptides and how do the response curves differ in MS1 and MS2? - the author could plot response curves for some example peptides. Also could the authors estimate the impact of interferences on MS1 and MS2?

Our accuracy tests were carried out at a low-load of 1ng. So at to 1:1 mix the heavy and light peptides are each present at 500pg, which drops for the light channel to ~167pg at the 1:5 mix. Most of the quantified peptides are at the lower end of the detection limit sensitivity wise and if we plot the intensity distribution according to accuracy, it appears that it is primarily the lower end of the abundance range which drops in accuracy (Rebuttal Figure 6A-B), supporting our claim that it becomes difficult to quantify the these peptides when the input material is so low. We selected a few proteins to show in specific cases how the deviation from the true value increases as the signal intensity decreases(Rebuttal Figure 6B). Unfortunately, Spectronaut does not provide a channel specific S/N value, but as this is highly correlated with the produced abundance values, we hope these data sufficiently support that the drop in accuracy is predominantly caused by a low quality signal. As interference levels have been investigated by other studies, our main goal was to compare HRMS1 precursor level quantification with standard DIA MS2 fragment based quantification. Since we do not collect enough MS2s for identification in our HRMS1 schemes, it becomes difficult to assess interference in our optimized method and could only do so for our standard DIA runs, serving as a relevant reference point, but not being a major focus of this work. To ensure the reader is made aware of this, we now included references to a few studies that have looked into MS1 and MS2 level quantification for DIA type acquisition (<https://www.sciencedirect.com/science/article/pii/S1535947620326475> and <https://www.sciencedirect.com/science/article/pii/S153594762035088X?via%3Dihub#fig1>). We would really like to thank the reviewer for making this point, as it inspired an idea how we can use some of the produced quality parameters (e.g using S/N, peak shape and Intensity properties to set thresholds for quantification that is likely to be highly erroneous) to filter out poorly quantified peptides/proteins from our single-cell data in the future.

Rebuttal Figure 6. 2D density distribution of protein log₂FC differences from expected values with respect to the light channel abundance on MS1 **A**) and MS2 **B**) level. The standard DIA data was used to generate the figures, so that both quant levels could be directly compared, as they have the same number of points per peak. **C-D**) Selected proteins to visualize the drop in accuracy dependence on protein abundance.

-In general the authors make the accuracy benchmarks based on visually comparing the distributions (e.g. Figure 2F). Could the authors instead do a more quantitative comparison (e.g. comparing offsets of ratios and/ or distribution widths)

Based on the reviewer's request, we have calculated the difference between measured log₂ fold-changes and expected values. As previously, we have plotted the distributions of these errors into a density plot where the upper part is based on HRMS1 precursor-level quant, standard DIA MS2 in the bottom. We can clearly see that MS1 quantification, when working with such limited input, is highly comparable to MS2, which was our overall goal of the experiment. To quantify these observations, we now calculate the full-width at half maximum (FWHM) of the error peaks and observe that HRMS1 appears to deliver quant values closer to the expected ones than MS2-DIA. We have substituted the original 2F figure with this one and moved the former 2F to the supplement. We have also added scatter plots, so that the error dependency on abundance can be appreciated.

Figure 2F. Density plots showing the deviation of the measured log₂FC from the expected values. Full-width at half maximum (FWHM) are provided next to the peaks.

Figure S3F. Scatter plot where the measurement error is plotted on the y-axis and the log₂ transformed heavy protein abundance on the x-axis. Left side represent HRMS1-DIA and right standart DIA quantification.

Lines: 345-353 “Interestingly, when higher ratio mixtures were compared, there appeared a minor, but clear discrepancy in MS2 level quantification, while MS1 ratio distribution remained centered around the expected value (Figure S3D). Higher MS1 accuracy was also observed comparing MS1 and MS2 protein ratios from the standard DIA method (Figure S3E). To provide a more quantitative accuracy comparison we evaluated the quantification error distributions at full-width at half maximum (Figure 2F). We could note that MS1 quantification leads to narrower distribution compared to MS2, for such low-input samples. Taken together, we conclude that WISH-DIA

enhances proteome depth from low-input samples while maintaining robust quantitative accuracy.”

-334 “Retention times were very robust across runs, with almost all precursor elution apex deviations being limited to 2.5 seconds (Figure 3E), underlining the solid chromatographic performance of this novel uPAC Neo Low Load column” Could the authors put this into context. E.g. putting a quantitative value on the variation and comparing it to other columns/technologies? (e.g. data from literature or in-house?)

In order to set this into perspective, we compared retention time (RT) stability with the U3000-uPAC and EvoSep-PepSep LC setups. We plotted the deviation for a single replicate (the middle one $n = 2$) from the RT deviation from the mean of 3 replicates. One can see clear shifts along the activity gradient with PepSep analytical column, while the RT with the uPAC are stably centered around 0 throughout the gradient. This figure has been incorporated into supplemental Figure S4F.

Figure S4F. Comparison of retention time (RT) stability with EvoSep-PepSep and U300-uPAC LC set-ups. The uPAC single-column or precolumn configuration is noted above the plots. The deviation of a single replicate ($n=2$) from the mean RT time of 3 replicas is plotted on the x-axis and the peptide elution peak apex time on the y axis. The different LC setups are indicated by color. Data for 1ng injection used in both comparisons.

Lines: 383-390 “To put the performance into perspective, we compared RT stability with our initially used column and observed a significant reduces RT fluctuations (Figure S4F), underlining the solid chromatographic performance of this novel uPAC Neo Low Load column.”

-Figure 3A: This Figure is out of context and I would remove it. I would also recommend to remove “micropillar array-based chromatography” from the title as the authors did not do much optimisation/benchmark based on their chromatography (see comment above)

We agree with the reviewer that in the context of the overall manuscript, it is out of place and have moved the figure to the supplement. We also revised the title of the manuscript, now that scope and focus of the manuscript has shifted since the first submission.

Lines: 1-3 “Exploration of cell state heterogeneity using single-cell proteomics through sensitivity-tailored data-independent acquisition”

-Figure 3D: could the authors comment on the 2 peaks in Figure 3D. Is that related with how the FWHM is calculated? Could the authors give more details on how such parameters have been calculated and could cause this.

Similarly to the Evosep platform the bimodality arises due to peptide elution profiles. We took the 45min method profile here as an example. In contrast, both peak populations are distributed continuously along the active gradient (Rebuttal Figure 7). We selected a few precursors to show specific examples of this (Rebuttal Figure 8). The FWHM values are taken directly from the Spectronaut software, which are calculated as the half maximum of the full peak based on the XIC. Together with Rebuttal Figure 1, this suggests that the peak calculations by the software reflect the actual profiles present in the raw data.

Rebuttal Figure 7. 2D density distribution of precursors peak full-width half maximum and elution apex time for the 45min uPAC LC method.

Rebuttal figure 8. Chromatogram showing wide and narrow peaks of precursor ions. The complete chromatogram is shown in the top and four select precursors are shown below (2 wide and 2 narrow).

-Figure S4 B legend: “Barplots showing numbers of proteins with single-column and pre-column configuration.” Could the authors label the Figures accordingly so that it is clear which numbers are related to pre-column and which to single column. Also it says Spectronaut16 “pre-release” instead of Spectronaut17 in the legend.

Labels have now been added above the plots in the main and supplemental figures. The legend has been corrected, thanks for pointing out these shortcomings.

-Figure 4 legend: seems wrong e.g. B “The same as A, but with the uPAC pre-column in-line” does not correspond to Figure 4B. Also C, D and E labels seem not to match the corresponding figure

Apologies for the mixed up figures legend. It has now been corrected

-Line 434: “This was primarily driven by the addition of a large number of proteins with low quantitative accuracy (Figure 5B).” I do not understand the relation to Figure 5B here. This needs to be better explained.

-Line 435: “The quantitative accuracy of the proteins that could be identified by directDIA remained similar, however some detrimental effects due to the library search can be noted, likely due to the detection of additional low-abundant peptides (Figure S5).” This benchmark is very qualitative and the “detrimental effect” needs more explanation as this is not clear from Figure S5.

We have elaborated on this section to hopefully clear up the loss of accuracy issue with high-load library searching:

Lines: 476-482 “To gain a better understanding of how the increased proteome coverage was affecting the overall quantitative accuracy of the data, we extracted the proteins that could be identified with directDIA or only with the implementation of a HL or GPF library and re-plotted the ratio distributions (Figure 4B). The additionally quantified proteins of those HL or GPF searches compared to directDIA alone specifically had a strikingly wider distribution, indicating significantly

increased deviation from the true values on those additionally identified peptides and proteins. (Figure S6). “

-Line 417: “However, the exact impact of using.. are potentially lost during single-cell processing might be erroneously quantified.” This statement would indicate that the FDR control is wrong- could the author comment on this or modify the statement accordingly.

We apologize that we gave the impression that we questioned the FDR control. Instead, we wanted to highlight that potentially these extra IDs have a relatively poor signal-to-noise, especially in terms of their quantitation, which would limit their usability when trying to explore biological variation. We have rephrased the sentence accordingly:

Lines: 459-461 “However, the exact impact of using such high-load (HL) ID transfer approaches remains unclear, especially in terms of quantification accuracy and consequently, biological information captured by the additional proteome coverage.”

-Figure 6B: QBL is mentioned in the legend but not explained. Figure needs more explanation in legend and/or text. In general I would recommend expanding Figure legends as some Figure legends have very little information.

Thanks for picking up this discrepancy, which we have now corrected. Originally, we referred to gas-phase fractionation (GPF) as quadrupole-based fractionation (QBL), but quickly learned that the broadly accepted term is GPF which was then adopted. Overall, we tried to expand on all the figure legends and hope they will ease the interpretation of the displayed data.

-Line 486: “...(PCA) and non-linear (UMAP) methods to gauge this biological variation (Figure 5D).” This should be Figure 6D? Same in line 492

Thank you, this is now corrected

Reviewer #3 (Remarks to the Author):

Petrosius et al. describes an analysis of orbitrap-based data-independent acquisition (DIA) for ultra-low-input samples. They describe a difference in optimal yield conditions between high- and low-input samples, and go on to recommend a protocol using standardized lab equipment that they claim is optimal for single-cell proteomics. The presented techniques are not novel, but the optimization is. The presented protocol seems timely and useful. I have but two minor comments:

* I found it hard to follow the difference between HRMS1 and WISH-DIA. It seems like HRMS1 is a good enough term for the protocol, as it is just a special case.

Although we initially refrained from generating an acronym for our method as we agree it to conceptually be a rather straightforward modification, having to refer to the method as wide-window-HRMS1-DIA became quite cumbersome. For simplicity's sake, WISH-DIA is easier to pronounce and present in talks and to collaborators, but we always ensure to clarify the link to HRMS1 and those hugely important early developments.

* It is hard to spot the differences in the presented violin plots. It would be easier to see reported differences in e.g. Figure 2F, and Figure A&B if scatterplots were used.

We appreciate the suggestion for adding the scatterplots. Besides several changes made in response to reviewers #1 and #2, we now also combine the density plots for accuracy with additional plots that show the overall differences/errors from the expected values and hope it will help with the clarity of the figure. To improve the clarity of our manuscript, so that everyone could appreciate the accuracy test we carried out we have also added the suggested scatter plot to the supplement, Figure S3F. We hope the reviewer agrees that in general, the plots are now easier to follow.

Figure S3F. Scatter plot where the measurement error is plotted on the y-axis and the log₂ transformed heavy protein abundance on the x-axis. Left side represent HRMS1-DIA and right standart DIA quantification.

REVIEWERS' COMMENTS

Reviewer #2 (Remarks to the Author):

The authors did a comprehensive revision and have addressed my comments. I recommend for publication.

Minor comment: Figure 7- Label E is missing.